Molecular Biology and Physiology

# From Modules to Networks: a Systems-Level Analysis of the Bacitracin Stress Response in *Bacillus subtilis*

Hannah Piepenbreier,[a] Andre Sim,[a] Carolin M. Kobras,[b] Jara Radeck,[c] Thorsten Mascher,[c] Susanne Gebhard,[b] Georg Fritz[a]*

[a]LOEWE Center for Synthetic Microbiology and Department of Physics, Philipps-Universität Marburg, Marburg, Germany
[b]Department of Biology & Biochemistry, Milner Centre for Evolution, University of Bath, Bath, United Kingdom
[c]Institute of Microbiology, Technische Universität Dresden, Dresden, Germany

**ABSTRACT** Bacterial resistance against antibiotics often involves multiple mechanisms that are interconnected to ensure robust protection. So far, the knowledge about underlying regulatory features of those resistance networks is sparse, since they can hardly be determined by experimentation alone. Here, we present the first computational approach to elucidate the interplay between multiple resistance modules against a single antibiotic and how regulatory network structure allows the cell to respond to and compensate for perturbations of resistance. Based on the response of *Bacillus subtilis* toward the cell wall synthesis-inhibiting antibiotic bacitracin, we developed a mathematical model that comprehensively describes the protective effect of two well-studied resistance modules (BceAB and BcrC) on the progression of the lipid II cycle. By integrating experimental measurements of expression levels, the model accurately predicts the efficacy of bacitracin against the *B. subtilis* wild type as well as mutant strains lacking one or both of the resistance modules. Our study reveals that bacitracin-induced changes in the properties of the lipid II cycle itself control the interplay between the two resistance modules. In particular, variations in the concentrations of UPP, the lipid II cycle intermediate that is targeted by bacitracin, connect the effect of the BceAB transporter and the homeostatic response via BcrC to an overall resistance response. We propose that monitoring changes in pathway properties caused by a stressor allows the cell to fine-tune deployment of multiple resistance systems and may serve as a cost-beneficial strategy to control the overall response toward this stressor.

**IMPORTANCE** Antibiotic resistance poses a major threat to global health, and systematic studies to understand the underlying resistance mechanisms are urgently needed. Although significant progress has been made in deciphering the mechanistic basis of individual resistance determinants, many bacterial species rely on the induction of a whole battery of resistance modules, and the complex regulatory networks controlling these modules in response to antibiotic stress are often poorly understood. In this work we combined experiments and theoretical modeling to decipher the resistance network of *Bacillus subtilis* against bacitracin, which inhibits cell wall biosynthesis in Gram-positive bacteria. We found a high level of cross-regulation between the two major resistance modules in response to bacitracin stress and quantified their effects on bacterial resistance. To rationalize our experimental data, we expanded a previously established computational model for the lipid II cycle through incorporating the quantitative action of the resistance modules. This led us to a systems-level description of the bacitracin stress response network that captures the complex interplay between resistance modules and the essential lipid II cycle of cell wall biosynthesis and accurately predicts the minimal inhibitory bacitracin concentration in all the studied mutants. With this, our study highlights how bacterial resistance emerges from an interlaced network of redundant homeostasis and stress response modules.

**Editor** Domitilla Del Vecchio, MIT

**Ad Hoc Peer Reviewer** Rushina Shah, MIT

This article followed an open peer review process. The review history can be read here.

Address correspondence to Georg Fritz, georg.fritz@uwa.edu.au.

* Present address: Georg Fritz, School of Molecular Sciences, The University of Western Australia, Perth, Western Australia, Australia.

**KEYWORDS** cell wall antibiotic, antimicrobial peptide, bacitracin, peptidoglycan, resistance network, regulatory network, computational model, antibiotic resistance

Computational approaches significantly improved our understanding of bacterial responses to environmental conditions, which often comprise multiple interconnected modules orchestrated in complex regulatory networks. For instance, mathematical modeling elucidated differences in signaling and signal processing in bacterial chemotaxis in *Bacillus subtilis* and *Escherichia coli* (1, 2), contributed to our understanding of how environmental and cellular conditions shape the complex phosphorelay system controlling sporulation and competence in *B. subtilis* (3–5), and helped to uncover the regulatory mechanisms of $\sigma^F$-dependent sporulation control in *Bacillus subtilis* (6, 7). In all of these studies, the overall cellular response toward environmental changes was shown to involve an intricate interplay between different regulatory modules, which can hardly be understood without theoretical frameworks.

The cell envelope stress response (CESR) is another example of a particularly important, multilayered regulatory network in bacteria, as it provides effective protection against crucial cell wall-targeting antibiotics, including the antimicrobial peptides (AMPs) bacitracin (BAC), ramoplanin, and vancomycin. In many bacteria, the CESR involves orchestrated expression of various resistance determinants that protect against these AMPs via an array of mechanisms (8). These include, for instance, changes in cell envelope composition to shield cellular targets from AMPs (9), production of resistance pumps to remove AMPs from their site of action (10), enzymatic or genetic modifications of target structures to prevent AMP binding (11), or the synthesis of immunity proteins to degrade AMPs altogether (12). Although many of the resistance mechanisms are well described and we have a good understanding of the gene regulatory control of individual resistance modules, the complex interplay and cross-regulation between individual resistance modules remain poorly understood. Given that 8 out of the 12 bacterial pathogens on the WHO's priority list have acquired resistance toward cell wall-targeting antibiotics ([https://www.who.int/news-room/detail/27-02-2017-who-publishes-list-of-bacteria-for-which-new-antibiotics-are-urgently-needed](https://www.who.int/news-room/detail/27-02-2017-who-publishes-list-of-bacteria-for-which-new-antibiotics-are-urgently-needed)), theoretical models rationalizing the cellular response toward such drugs are urgently needed.

To address this knowledge gap, we focused in this study on the resistance network of *B. subtilis* toward BAC, an AMP that interferes with the lipid II (LII) cycle of cell wall biosynthesis (Fig. 1) (13). Briefly, within this essential pathway the peptidoglycan (PG) precursors *N*-acetylglucosamine (GlcNAc) and *N*-acetylmuramic acid (MurNAc)-pentapeptide are sequentially attached to the lipid carrier molecule undecaprenyl phosphate (UP) by MraY and MurG, thereby forming lipid II (Fig. 1). Subsequently, lipid II is flipped across the cytoplasmic membrane via the flippases MurJ and Amj, where the PG monomer (GlcNAc-MurNAc-pentapeptide) is incorporated into the growing cell wall by various redundant penicillin-binding proteins (PBPs). This leaves the lipid carrier in its pyrophosphate form (UPP), which has to be recycled to UP by dephosphorylation (in *B. subtilis* via the UPP phosphatases BcrC, UppP, and, to a minor degree, YodM) (14–16) to allow a new round of PG monomer transport. Bacitracin blocks the cycle by forming a tight complex with UPP (UPP-BAC), which efficiently prevents recycling of the lipid carrier and ultimately leads to lysis of cells (17, 18). Like in many Gram-positive bacteria, bacitracin resistance in *B. subtilis* is mediated by multiple resistance determinants, which are transcriptionally upregulated in response to bacitracin treatment (reviewed in reference 19). The most effective (primary) resistance determinant is the ABC transporter BceAB (20), which protects UPP from the inhibitory grip of bacitracin (Fig. 2A)—presumably by breaking UPP-BAC complexes and thereby shifting the binding equilibrium toward the free form of UPP (21). The second line of defense is mediated by the UPP phosphatase BcrC, which increases the rate of UPP dephosphorylation and thereby promotes progression of the lipid II cycle (14, 15, 22, 23) (Fig. 2A). Simultaneously, *B. subtilis* induces production of the phage shock-like proteins LiaI and

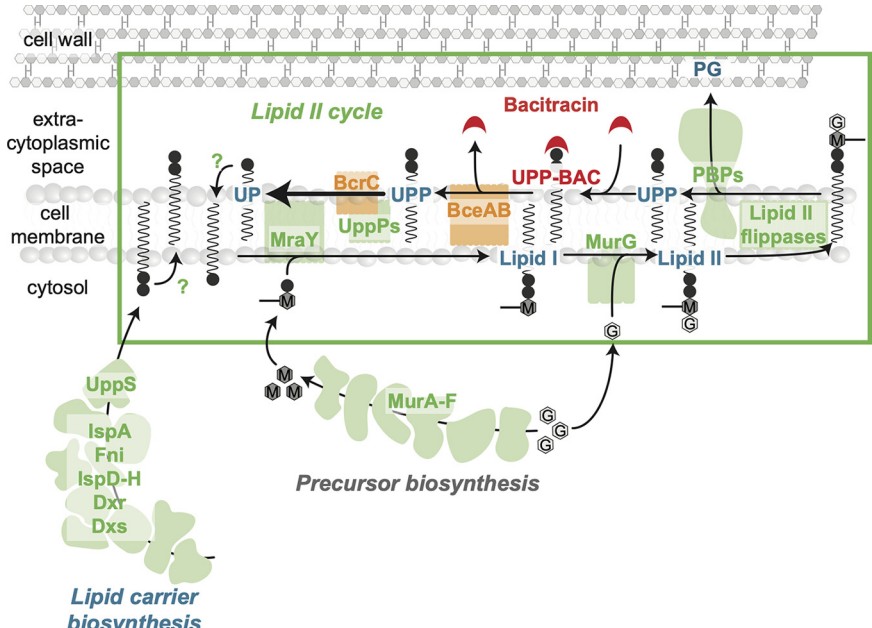

**FIG 1** Scheme of the cell wall biosynthetic pathway and its inhibition by bacitracin. The lipid II cycle of cell wall biosynthesis is responsible for the translocation of PG precursors across the cytoplasmic membrane and represents the rate-limiting step in this process. The cytoplasmic production of UDP-MurNAc-pentapeptide (M) from UDP-GlcNAc (G) is catalyzed by the MurA to -F ligases. Subsequently, at the internal leaflet of the cytoplasmic membrane the translocase MraY and the transferase MurG sequentially attach UDP-MurNAc-pentapeptide and UDP-GlcNAc to the lipid carrier undecaprenyl phosphate (UP), giving rise to the lipid I and lipid II intermediates, respectively. Various flippases translocate lipid II to the outer leaflet of the cytoplasmic membrane, where penicillin-binding proteins (PBPs) incorporate the subunits into the growing PG layer. In order to recycle the resulting pyrophosphorylated state of the lipid carrier (UPP), dephosphorylation by UPP phosphatases (UppPs), including BcrC, yield the initial substrate UP for another round of PG subunit transport. Lipid carrier recycling requires flipping of UP to the internal leaflet by a yet-unknown mechanism. Finally, dilution of lipid carriers is counterbalanced by cytoplasmic synthesis of UPP, which involves the isoprenoid biosynthesis pathway with the undecaprenyl pyrophosphate synthetase UppS catalyzing the last committed step. Like for UP flipping, the required mechanism to present UPP to the externally acting phosphatases is unknown. Bacitracin inhibits the lipid II cycle by binding to UPP, thereby preventing UPP dephosphorylation and progression of the cycle.

LiaH (22, 24), which play only a minor role in bacitracin resistance and seem to be involved in stabilization of membrane integrity by a mechanism that is yet to be determined (25, 26).

Expression of *bceAB* is activated by a two-component system comprising the histidine kinase BceS and the response regulator BceR (15, 20, 22, 27, 39). BceS forms a sensory complex with BceAB in the membrane (28, 29), which acts as a "flux sensor" reporting on the antibiotic load experienced by each individual transporter—thereby activating further transporter expression only if their detoxification capacity approaches saturation (21). Expression of BcrC is primarily controlled by the extracytoplasmic function sigma factor $\sigma^M$ (14, 30). While the physiological input triggering activation of $\sigma^M$ still remains elusive (31, 32), the broad range of inducing conditions, including cell wall antibiotics, salt, ethanol, and others, suggests that it is not a specific chemical compound but rather a cellular cue upon cell envelope damage that activates the $\sigma^M$ response (33). Interestingly, despite the seemingly unrelated input stimuli for the BceAB and the BcrC resistance modules—with BceAB being activated by a "drug-sensing" mechanism (antibiotic flux) and BcrC by a "damage-sensing" mechanism— previous work revealed that there is a high level of interdependency between the modules (26). In particular, in mutants lacking the ABC transporter BceAB, the secondary layers of resistance are induced more strongly and *vice versa*, suggesting that this compensatory regulation is the origin of robust cell wall homeostasis in *B. subtilis*.

In this study, we set out to decipher the regulatory interplay between these two

## A  Theoretical model

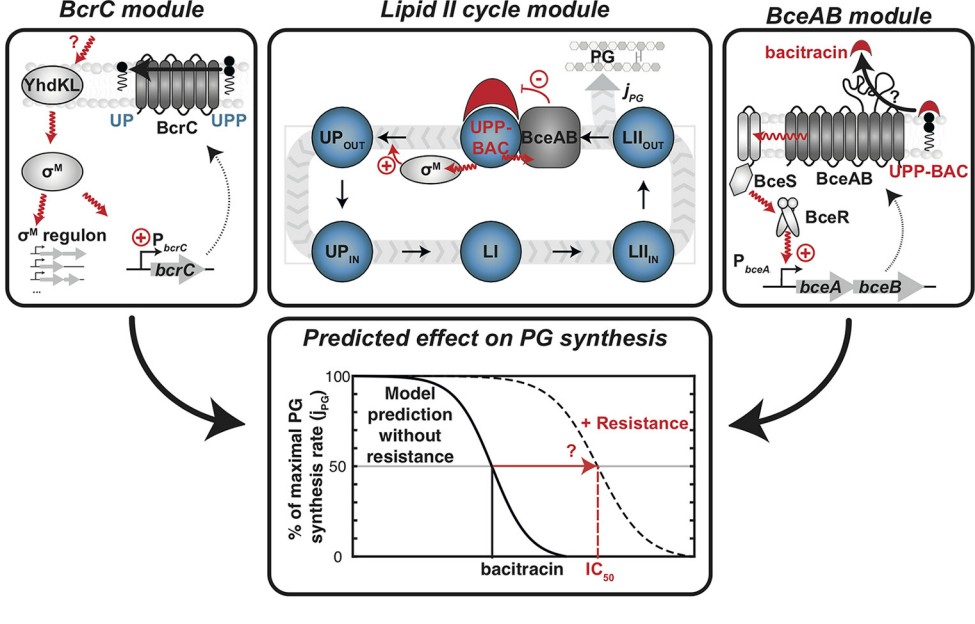

## B  Experimental analysis

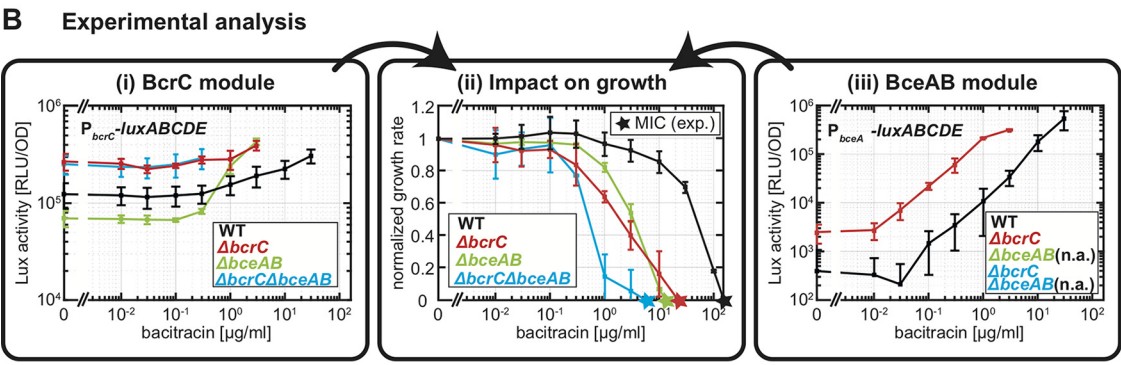

**FIG 2** Modular composition of the bacitracin stress response network and its experimental analysis in *B. subtilis*. (A) Our theoretical model of the bacitracin stress response network is based on three interconnected modules. At the core of the model is a previously established theoretical description of the lipid II cycle module (center diagram), which predicts the PG synthesis rate ($j_{PG}$) of a *B. subtilis* strain devoid of any inducible resistance determinants under antibiotic perturbation (34). The dynamic variables within the model are the concentrations of the lipid II cycle intermediates (blue spheres) in the inner and outer leaflet of the cytoplasmic membrane, as indicated by the subscripts IN and OUT. To arrive at a model for wild-type cells, we first incorporated the action of the $\sigma^M$ module (left diagram), in which an unknown cue activates the anti-$\sigma$ factors YdhL and YhdK in response to cell envelope stress, triggering the release of $\sigma^M$ and the concomitant upregulation of BcrC and an array of further $\sigma^M$-dependent genes. As a second module we incorporated the action of the ABC transporter BceAB (right diagram), in which a complex of the histidine kinase BceS and the BceAB transporter jointly act as a sensor for bacitracin flux, triggering phosphorylation of the response regulator BceR and concomitant upregulation of *bceAB* expression. Both resistance modules are qualitatively expected to increase resistance (lower diagram) by shifting the bacitracin concentration at which the PG synthesis rate reaches its half-maximal level (IC$_{50}$) to higher values. (B) Experimental analysis of resistance module gene expression and their impact on the growth rate. Shown are target promoter activities of P$_{bcrC}$-*luxABCDE* (i) and P$_{bceA}$-*luxABCDE* (iii) in *B. subtilis* strains carrying indicated deletions of resistance modules, as given by specific luciferase activity (RLU/OD$_{600}$) 1 h after addition of indicated amounts of bacitracin. Panel ii shows the corresponding normalized growth rates, which are obtained as the average growth rates of strains carrying the different reporter constructs but sharing the same genotype. Measurements were performed during exponential growth phase in LB medium at 37°C in a microtiter plate reader. Data are shown for strains TMB1619 (wild-type strain W168 *sacA*::pCHlux103 [P$_{bceA}$-*lux*]), TMB1620 (wild-type strain W168 *sacA*::pCHlux104 [P$_{bcrC}$-*lux*]), TMB1623 (W168 *bceAB*::kan *sacA*::pCHlux103 [P$_{bceA}$-*lux*]), TMB1624 (W168 *bceAB*::kan *sacA*::pCHlux104 [P$_{bcrC}$-*lux*]), TMB1627 (W168 *bcrC*::tet *sacA*::pCHlux103 [P$_{bceA}$-*lux*]), TMB1628 (W168 *bcrC*::tet *sacA*::pCHlux104 [P$_{bcrC}$-*lux*]), and TMB1632 (W168 *bceAB*::kan *bcrC*::tet *sacA*::pCHlux104 [P$_{bcrC}$-*lux*]) (Table S1). Data points and error bars indicate means and SDs derived from at least three biological replicates. Note that we did not test P$_{bceA}$-*luxABCDE* activity in strains carrying a Δ*bceAB* deletion, because the flux-sensing mechanism activating P$_{bceA}$ strictly relies on the presence of BceAB (21). The colored stars indicate the experimental MIC values, calculated as the concentration at which a linear interpolation between the data points crosses the zero line. In case of the wild-type strain, the MIC was calculated by a linear extrapolation to the zero line, given that the highest concentration tested did not fully inhibit growth. n.a., not applicable.

resistance determinants by considering the dynamics of the lipid II cycle as a pivotal connection between drug- and damage-sensing resistance modules. To this end, we took advantage of a recently established computational model for the lipid II cycle, which describes the dynamics of PG synthesis based on biochemical parameters of the involved enzymes and cycle intermediates (34). By integrating the existing mathematical description of the lipid II cycle with a previously established model for BceAB resistance module (21) and a novel theoretical description of the BcrC module (Fig. 2A), we developed a systems-level description of the bacitracin stress response that captures the MIC of a wild-type (WT) *B. subtilis* strain, as well as several mutants deleted for resistance systems individually or in combination. Additionally, our theoretical and experimental analyses reveal that an increased total number of lipid carriers, i.e., the sum of UP, UPP, lipid I, and lipid II, and an accumulation of the lipid carrier UPP in a Δ*bcrC* mutant are the origin of the significantly higher impact of the BceAB resistance module on bacitracin resistance when BcrC is lacking. As our model does not include any additional layers of regulation, our results show that the properties of the lipid II cycle itself contribute to the homeostatic control of the overall resistance response toward bacitracin. Thus, the theory presented here not only provides a comprehensive quantitative description of the bacitracin resistance network in *B. subtilis* but also uncovers regulatory mechanisms of the multilayered response toward this antibiotic.

## RESULTS

The common purpose of the UPP phosphatase BcrC and the ABC transporter BceAB is to ensure progression of the lipid II cycle under bacitracin treatment, since bacitracin inhibits an important step of this cycle (Fig. 2A). Consequently, to study the impact of the two resistance modules on lipid II cycle homeostasis, we successively integrated the resistance modules into a detailed computational description of the lipid II cycle (34). This previously established theory predicts the rate of peptidoglycan synthesis ($j_{PG}$) of a *B. subtilis* strain devoid of any inducible resistance determinants under antibiotic perturbation (Fig. 2A, upper central diagram) and thus served as the basis of our mathematical model of the overall bacitracin resistance network. In the first step, we included the BcrC module and studied its protective effect on the lipid II cycle (Fig. 2A, left diagram). In a second step, we investigated the interaction of the two resistance modules by integrating the preexisting theory of the BceAB module (21) into the model (Fig. 2A, right diagram). In this full model, we were able to study the impact of the two modules on bacitracin resistance by quantifying the shift in the bacitracin concentration at which the PG synthesis rate reaches its half-maximal level ($IC_{50}$) to higher values (Fig. 2A, lower diagram).

**Impact of the UPP phosphatase BcrC on bacitracin resistance.** In our previous computational description of the lipid II cycle (34), we made the simplifying assumption that the enzymes involved in lipid II cycle progression feature constant expression levels under antibiotic treatment, but it is known that the UPP phosphatase BcrC is upregulated in response to bacitracin treatment. Also, the model did not include the activity of the BceAB transporter and therefore was only able to predict the approximate MIC for bacitracin in a Δ*bceAB* mutant strain of *B. subtilis* (34). Thus, the first step in arriving at a more realistic description of lipid II cycle homeostasis was to include the bacitracin-dependent upregulation of *bcrC* expression into our computational model for the Δ*bceAB* mutant strain. To experimentally assess *bcrC* expression in response to bacitracin treatment under our experimental conditions, we integrated a P$_{bcrC}$-luxABCDE reporter construct into the chromosome of a Δ*bceAB* mutant and measured luciferase activity 1 h after addition of various bacitracin levels (Fig. 2Bi, green). The P$_{bcrC}$ promoter activity clearly correlated with increasing levels of bacitracin, leading to a maximal ~6-fold induction at 3 μg/ml of bacitracin compared to that under the untreated condition. In contrast, wild-type cells displayed only an ~3-fold P$_{bcrC}$ induction reached at 10-fold-higher bacitracin levels (30 μg/ml) (Fig. 2Bi, black), suggesting that the additional expression of *bceAB* in the wild type mitigates the demand for *bcrC* expression, as discussed further below. Furthermore, we investigated the impact of the

mSystems®

BcrC resistance module on bacitracin resistance by comparing the growth of a Δ*bceAB* mutant and a Δ*bceAB* Δ*bcrC* double mutant. In doing so, we were able to study the resistance contribution of BcrC alone and avoided any compensatory upregulation of *bceAB* expression that may complicate the interpretation of a comparison between the wild type and a Δ*bcrC* mutant strain. By defining the MIC as the lowest antibiotic concentration leading to zero growth rate after bacitracin addition, we observed a 2.3-fold-lower MIC value for the Δ*bceAB* Δ*bcrC* mutant (Fig. 2Bii, light blue star; MIC$^{ΔbceABΔbcrC}$, ~6.3 μg/ml) compared to the Δ*bceAB* mutant (Fig. 2Bii, green star; MIC$^{ΔbceAB}$, ~14.5 μg/ml), consistent with earlier results (26). This clearly confirmed that the BcrC resistance module by itself contributes significantly to the growth of *B. subtilis* under bacitracin treatment.

Next, we incorporated the observed upregulation of *bcrC* into our existing model of the lipid II cycle, with the goal of accurately predicting the antibiotic susceptibility toward bacitracin in the Δ*bceAB* mutant strain. Briefly, the previous model of the lipid II cycle (34) considered Michaelis-Menten kinetics for all characterized enzymes, while the mostly unknown flipping reactions of the intermediates UPP, UP, and lipid II were described by first-order kinetics (see Text S1 in the supplemental material for a detailed description of the model). To integrate the different levels of *bcrC* expression in response to bacitracin into our model, we modified the mathematical description of the dephosphorylation reaction of UPP. Since the speed of every enzymatic reaction within the lipid II cycle is proportional to the concentration of the enzymes that catalyze the reaction—according to Michaelis-Menten theory—the bacitracin-induced increase in BcrC levels implies an increase in the speed of UPP dephosphorylation (Fig. 3A and C). However, in *B. subtilis* the dephosphorylation of UPP is additionally catalyzed by a second phosphatase, UppP, the expression of which is independent of bacitracin (14–16) (Fig. 3B). Thus, the total speed of the UPP dephosphorylation reaction is proportional to the weighted sum of the bacitracin-dependent contribution from BcrC and the bacitracin-independent contribution from UppP (Fig. 3C), as indicated in equation 1.

$$\text{rate of UPP dephosphorylation} \sim x^{BcrC} \times f^{BcrC}(\text{BAC}) + [1 - x^{BcrC}] \times 1 \qquad (1)$$

Here, the factor $x^{BcrC}$ quantifies the fractional contribution of BcrC and $1 - x^{BcrC}$ the fractional contribution of UppP to the total phosphatase activity in the absence of bacitracin. Moreover, the upregulation of BcrC levels under bacitracin treatment leads to a fold induction of the UPP phosphatase activity according to the factor $f^{BcrC}$ (BAC), which ranges from 1 to a maximal fold change (Fig. 3A). Thus, the stronger the contribution of BcrC toward the overall phosphatase activity (higher $x^{BcrC}$), the more pronounced the acceleration of the UPP dephosphorylation reaction in response to bacitracin (Fig. 3C). To determine the unknown parameter $x^{BcrC}$, we first assumed that the BcrC protein level is proportional to the detected luminescence output from the P$_{bcrC}$-*luxABCDE* reporter in the *bceAB* mutant (Fig. 2Bi). We then simulated the model of the lipid II cycle for different values of $x^{BcrC}$ and monitored how the upregulation of BcrC in response to bacitracin affected the overall rate of PG synthesis (Fig. 3D). Here, it turned out that in the case of negligible impact of BcrC to the total phosphatase activity ($x^{BcrC} = 0$), the UPP phosphatase remains constant for all bacitracin concentrations, such that the PG synthesis rate decreases hyperbolically with increasing bacitracin concentration, as seen before (34). In contrast, for high values of $x^{BcrC}$, i.e., in the case that the total phosphatase activity is dominated by BcrC, the upregulation of BcrC (and thus phosphatase activity) leads to a speed-up (and thus to a peak) of the PG synthesis rate at intermediate bacitracin levels. Ultimately, this shifts the IC$_{50}$ value— the antibiotic concentration reducing the PG synthesis rate to 50% of its unperturbed rate—to a higher value. Studying the dependence of the predicted IC$_{50}$ values on different values of $x^{BcrC}$ showed that the IC$_{50}$ matched the experimentally measured MIC of the Δ*bceAB* mutant when UppP and BcrC made approximately equal contributions to the overall phosphatase activity (Fig. 4A).

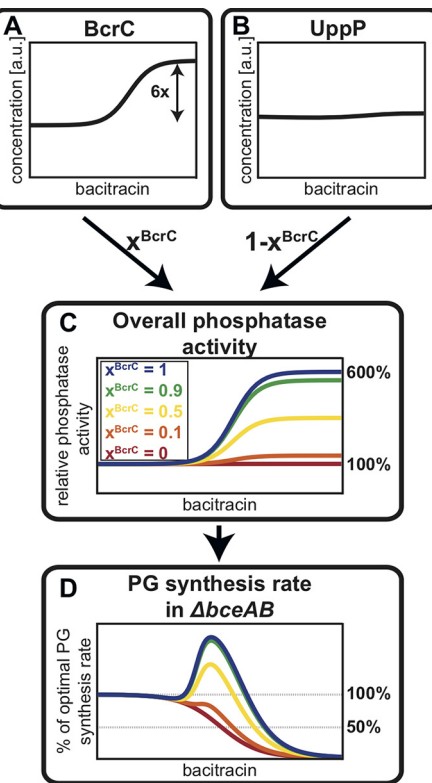

**FIG 3** Different contributions of BcrC and UppP to the overall UPP phosphatase activity lead to variable levels of protection against bacitracin. To capture the influence of the BcrC module to lipid II cycle homeostasis, the bacitracin-dependent induction profiles of the two phosphatases corresponding to *bcrC* (A) and *uppP* (B) were used as proxies for their contributions to the total UPP phosphatase activity. Given that no biochemical characterization regarding the relative phosphatase activities of the two proteins exists, we introduce the parameter $x^{BcrC}$ describing the relative contribution of BcrC (and $1 - x^{BcrC}$ the contribution of UppP) to the overall phosphatase activity in the absence of bacitracin. The bacitracin-dependent overall phosphatase activity varies according to these impacts of BcrC and UppP, as illustrated in panel C. Integrating the bacitracin-dependent UPP phosphatase activity in panel C in the model for the lipid II cycle (34) leads to predictions for the PG synthesis rate in panel D. This shows that the stronger the contribution by $x^{BcrC}$, the higher the bacitracin concentration at which the PG synthesis rate reaches its half-maximal value, which we define as the $IC_{50}$. Previous work showed that the $IC_{50}$ serves as a good proxy for the experimental MIC for various cell wall antibiotics (34).

**The total amount of lipid carrier is increased under *bcrC* deletion to ensure a close-to-optimal PG synthesis rate.** If the two phosphatases make approximately equal contributions to the overall phosphatase activity, one would predict that the deletion of one of the phosphatases should have significant impact on cellular physiology even in the absence of cell wall antibiotics. In fact, within our model the deletion of *bcrC* significantly reduces the speed of the dephosphorylation reaction of the lipid II cycle, leading to an accumulation of UPP compared to that in a model with wild-type *bcrC* activity (Fig. 5A and B). Given that the lipid II cycle can be approximated as a closed-loop system in which the total amount of lipid intermediates running through the cycle stays constant (34), the accumulation of UPP simultaneously reduces the concentrations of all other lipid II cycle carriers (Fig. 5B). In addition, especially the shortage of lipid II directly leads to a distinct reduction of the overall rate of PG synthesis of the lipid II cycle, since lipid II ultimately releases the PG monomers for incorporation into the cell wall (see theoretical description of the PG synthesis rate in the supplemental material). Hence, within the simulated range of BcrC and UppP contributions to the overall phosphatase activity, the lack of BcrC was predicted to reduce the rate of PG synthesis below its half-maximum for some of the parameters tested. Furthermore, the model predicted relatively low $IC_{50}$ values since the overall PG

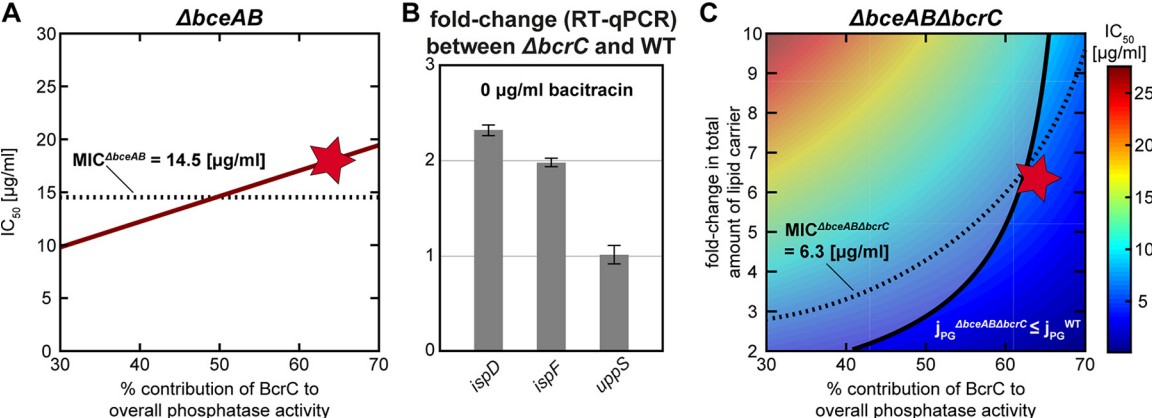

**FIG 4** Calibration of a model integrating the lipid II cycle with the BcrC resistance module. (A) Combining the lipid II cycle module with the BcrC module (cf. Fig. 2A) leads to a model describing a $\Delta bceAB$ mutant strain. Predictions of this model for the $IC_{50}$ value (the bacitracin concentration at which the PG synthesis rate declines to 50% of the unperturbed value) are shown under various contributions of BcrC ($x^{BcrC}$) to the overall UPP phosphatase activity (red line). The linear increase of the $IC_{50}$ value with $x^{BcrC}$ is the result of the stronger overall phosphatase activity incurred by BcrC upregulation (cf. Fig. 3C and D). The dotted line shows the experimental MIC of bacitracin in a $\Delta bceAB$ mutant, and the red star indicates the optimal parameter obtained by in our constrained optimization approach (see Text S1). (B) In the absence of bacitracin, expression of $ispD$ and $ispF$ is upregulated in a $\Delta bcrC$ mutant relative to the *B. subtilis* wild type, as quantified by RT-qPCR as described in Materials and Methods. Given that $ispD$ and $ispF$ are involved in early steps of UPP *de novo* synthesis, this suggests that the deletion of $bcrC$ triggers elevated levels of lipid II cycle intermediates, which may, in turn, compensate for the reduced UPP dephosphorylation rate in this mutant. (C) Predictions of the bacitracin $IC_{50}$ in a model for the $\Delta bceAB$ $\Delta bcrC$ double mutant (see color key) as a function of various contributions of BcrC to the overall UPP phosphatase activity (x axis) and the fold change of total lipid II cycle intermediates as induced by the $bcrC$ deletion (y axis). Within this model, the higher $x^{BcrC}$ in the model in the $\Delta bceAB$ mutant, the stronger the $bcrC$ deletion in the double mutant reduces the $IC_{50}$ value. Accordingly, in order to achieve a similar $IC_{50}$ value (same color in the background), higher $x^{BcrC}$ fractions require higher upregulation of the total abundance of lipid II cycle intermediates in this model. The dashed line indicates the experimental MIC of bacitracin in a $\Delta bceAB$ $\Delta bcrC$ double mutant. The parameters below the solid black line represent physiologically plausible combinations, in which the PG synthesis rate in the mutant ($j_{PG}^{\Delta bceAB\Delta bcrC}$) does not exceed the rate in the wild-type ($j_{PG}^{WT}$). The red star indicates the optimal parameter set obtained by in our constrained optimization approach (see Text S1).

synthesis rate was already reduced and very small amounts of bacitracin should be sufficient to further decrease it to the assumed critical rate of 50% of its optimal level.

However, such a reduction in the overall rate of PG synthesis even without bacitracin treatment would cause clear defects in cell growth, which was observable neither in the absence of bacitracin nor under low bacitracin concentrations when monitoring the growth of the $\Delta bceAB$ $\Delta bcrC$ mutant strain. Instead, the growth rate of the double mutant was only slightly affected without bacitracin (Fig. S1) and the experimentally determined MIC (Fig. 2Bii, light blue) was significantly higher than the $IC_{50}$ values predicted by the model. These results led us speculate that *B. subtilis* uses additional routes to respond to the deletion of $bcrC$—thereby ensuring a close-to-optimal rate of PG synthesis—probably by increasing the concentrations of lipid II. How does the cell implement this homeostatic control? As previous studies revealed, $\sigma^M$ not only regulates the expression of $bcrC$ in response to bacitracin but also induces individual steps of the methylerythritol phosphate (MEP) pathway (e.g., the $ispDF$ operon), which is responsible for early steps of lipid carrier (UPP) synthesis (30, 35). Indeed, it was shown that the expression of $\sigma^M$ itself is significantly increased in a mutant strain lacking BcrC (15). Hence, we hypothesized that the lipid II concentrations might be homeostatically regulated by $\sigma^M$-dependent control of the production of new lipid carrier. To test this hypothesis, we used reverse transcription-quantitative PCR (RT-qPCR) to quantify transcript levels of $\sigma^M$-regulated genes involved in the production of UPP synthesis. As illustrated in Fig. 4B, we found that during exponential growth in LB medium (without bacitracin), expression of both $ispD$ and $ispF$ was 2-fold higher in the $bcrC$ deletion strain than in the wild-type strain. Furthermore, the UPP synthetase encoded by $uppS$, which is not part of the $\sigma^M$ regulon, did not show differential expression between the two strains (Fig. 4B). These results suggest that the upregulation of early steps of UPP synthesis in a $bcrC$ deletion mutant may increase the overall abundance of lipid carriers in the lipid II cycle, thereby counteracting the bottleneck induced by UPP phosphatase deletion.

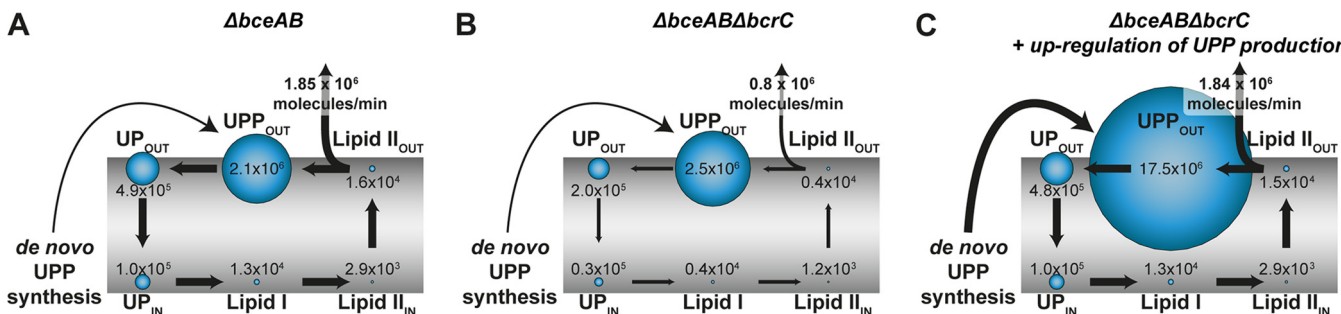

**FIG 5** Model-predicted distribution of lipid II (LII) cycle intermediates in the ΔbceAB and ΔbceAB ΔbcrC mutant strains. The distribution of the different lipid II cycle intermediates without bacitracin treatment is highly asymmetric. The lipid II cycle is located around the cell membrane, which is indicated in gray. Lipid II cycle intermediates are illustrated with blue circles, while the size of the circles correlates with the concentration of the respective intermediate. UPP$_{OUT}$, UP$_{OUT}$, and LII$_{OUT}$ represent the fraction the intermediates UPP, UP, and lipid II, respectively, which is located at the outer leaflet of the cell membrane. Accordingly, the fraction of intermediates located at the inner leaflet of the cell membrane is described by UP$_{IN}$ and LII$_{IN}$. UP$_{IN}$ is not displayed, as this lipid intermediate is not directly involved in the lipid II cycle. Lipid I (LI) is present solely on the inner leaflet of the cell membrane. The *de novo* synthesis of new lipid carrier in the form of UPP is indicated. The thickness of the arrows correlates with the fluxes from one intermediate state into the next one within the lipid II cycle. In addition, the rate of PG synthesis is displayed. (A) When BceAB is lacking (ΔbceAB), the concentrations of the various lipid II cycle intermediates equal the concentrations predicted in the basic model without bacitracin stress response determinants (34). While UPP$_{OUT}$ is most abundant, the concentrations of LI, LII$_{IN}$, and LII$_{OUT}$ are 2 orders of magnitude lower. UP$_{IN}$ and UP$_{OUT}$ are present in intermediate concentrations. (B) When *bcrC* is additionally deleted (ΔbceAB ΔbcrC), the rate of UPP dephosphorylation is significantly reduced and lipid intermediates accumulate in the form of UPP$_{OUT}$, as this is the substrate of the respective reaction. Since the lipid II cycle is a closed-loop system (34), all other concentrations are depleted concomitantly. However, as UPP$_{OUT}$ is still the most abundant intermediate in the lipid II cycle, its concentration is not raised significantly. The distinct reduction (>50%) of the concentrations of UP$_{OUT}$, UP$_{IN}$, LI, LII$_{IN}$, and LII$_{OUT}$ leads to significantly reduced fluxes within the lipid II cycle. In particular, the reduction of the concentration of LII$_{OUT}$ of ~75% leads to a decreased rate of PG synthesis, far below the half-maximal level. (C) However, the model predicts a nearly unaffected rate of PG synthesis when a 6.6-fold increase in total lipid intermediates (caused by a higher rate of UPP *de novo* synthesis) in response to *bcrC* deletion is expected. While the concentration of UPP$_{OUT}$ is massively increased, all other lipid II cycle intermediates are as abundant as in the ΔbceAB mutant scenario. Consequently, the similar concentrations of LII$_{OUT}$ imply similar rates of PG synthesis in both the mutant lacking BceAB exclusively (ΔbceAB) and the mutant lacking both resistance modules (ΔbceAB ΔbcrC).

Hence, we asked how a higher total concentration of the lipid II cycle intermediates (in the following referred to as LII intermediates) would affect the model prediction for a *bcrC* deletion mutant (see Text S1 for a detailed description). As we did not know the precise change in LII intermediates in the *bcrC* deletion strain, we simulated the model for different fold changes in LII intermediates and predicted the IC$_{50}$ under bacitracin treatment. In this study, it turned out that an increase in LII intermediates raises both the lipid II concentrations and the overall PG synthesis rate (Fig. 5C) and thus ensures progression of the lipid II cycle without bacitracin treatment, as suggested by the fact that growth of the *bcrC* deletion strain was not drastically reduced compared to that of the wild type (Fig. S1). Accordingly, the model predicts that higher levels of LII intermediates in the ΔbceAB ΔbcrC mutant lead to higher IC$_{50}$ values under bacitracin treatment (Fig. 4C, dependence along vertical axis). The model also predicts that the higher the contribution of BcrC to the overall phosphatase activity, the higher the required fold change in LII intermediates to reach the same IC$_{50}$ values in the ΔbceAB ΔbcrC mutant (Fig. 4C, dependence along horizontal axis). This underlines the idea that upregulation of LII intermediates can compensate for the lack of UPP phosphatase activity. However, we also noted that some of the tested parameter combinations led to predictions in which the PG synthesis rate in the mutant ($j_{PG}^{\Delta bceAB\Delta bcrC}$) was higher than the rate in the WT ($j_{PG}^{WT}$), which is physiologically implausible (Fig. 4C, shaded area). Thus, to arrive at a physiologically plausible parameter set, we performed parameter optimization to simultaneously fit the experimental MIC values of the ΔbceAB and ΔbceAB ΔbcrC mutant strains while meeting the constraint $j_{PG}^{\Delta bceAB\Delta bcrC} \leq j_{PG}^{WT}$. This resulted in a set of parameters in which BcrC was the dominant phosphatase ($x^{BcrC} = 63\%$ [±9.5%]) and in which the LII intermediate level was upregulated 6.2-fold (±0.7-fold), resulting in a close-to-optimal PG synthesis rate and IC$_{50}$ predictions ranging closely around the measured MICs for both mutant strains (Fig. 4A and C, red stars). These results are in line with previous experimental studies with *B. subtilis* (16) showing that a *bcrC* deletion led to a stronger reduction in resistance against bacitracin than a deletion of *uppP*. Hence, by integrating the homeostatic control of the overall LII intermediate level in response to a lack of BcrC, we arrived at a theoretical model that

quantifies the impact of BcrC as a secondary resistance module on the progression of the lipid II cycle, both in the absence and in the presence of bacitracin.

**Interaction between the BcrC and BceAB resistance modules.** Next, we focused on the interplay between the primary and secondary resistance modules BceAB and BcrC, respectively. As noted above, it turned out that the presence of BceAB in the wild type mitigates the demand for *bcrC* expression, as reflected in an ~2-fold-lower induction in $P_{bcrC}$ activity upon bacitracin treatment in the wild type than in the $\Delta bceAB$ mutant (Fig. 2Bi, black versus green line). We found that the $P_{bceA}$ promoter displays an ~10-fold-lower activity in the wild type compared to a $\Delta bcrC$ mutant, suggesting that also the presence of BcrC reduces the demand for *bceAB* expression in wild-type cells. These results clearly indicate a high level of cross-regulation between the resistance modules, which we wanted to rationalize via our computational model.

In order to complete the computational model of the bacitracin resistance network in the *B. subtilis* wild type, we next integrated the BceAB transporter into our theory of the lipid II cycle. To this end, we took advantage of a previously developed theoretical description of the BceAB resistance module in *B. subtilis* (21). Briefly, this model is based on differential equations describing the binding of bacitracin (BAC) to UPP (yielding UPP-BAC complexes), the upregulation of *bceAB* expression in response to increasing UPP-BAC complexes and ultimately the release of bacitracin from UPP catalyzed by increasing BceAB transporters levels. Here, the relative bacitracin flux ($J_{BAC}$) experienced by each BceAB transporter is described by Michaelis-Menten kinetics,

$$J_{BAC} = \frac{[\text{UPP-BAC}]}{K_m + [\text{UPP-BAC}]}$$

Where $K_m$ is the UPP-BAC concentration at which the catalytic rate of BceAB reaches its half-maximal value. Importantly, within this model $J_{BAC}$ not only sets the rate at which UPP is released from the inhibitory grip of bacitracin but also regulates expression of *bceAB* expression via the flux-sensing mechanism depicted in Fig. 2A and as detailed in Text S1. Combining this model for the BceAB module with the equations for the lipid II cycle, including the bacitracin-dependent upregulation of the BcrC module, ultimately led us to a comprehensive computational model for both resistance modules in the *B. subtilis* wild-type strain (see Text S1 for a detailed model description).

We then asked whether this full model is in quantitative agreement with the data describing the bacitracin response of the BceAB module in wild-type cells. To this end, we fixed all parameters for the lipid II cycle to the optimal values derived above and imposed that the regulation of BcrC protein levels is proportional to the $P_{bcrC}$ promoter activity measured in the wild-type strain (Fig. 2Bi, black data). Then we estimated the additional parameters describing the induction of the BceAB resistance module by a fit to the experimental $P_{bceA}$ promoter activity (Fig. 2Biii, black data; see Materials and Methods for details of the fitting procedure and Table S3 for parameter values). This led us to a parameter combination for which the model output closely resembles the observed response of $P_{bceA}$ toward bacitracin (Fig. 6A), suggesting that the model accurately captures the cross-regulation between the two resistance modules. Importantly, when studying the PG synthesis rate within this model (Fig. 6B, black line), it turned out that with increasing bacitracin concentration the increasing production of BcrC and BceAB stabilized the PG synthesis rate even in the presence of bacitracin and increased the $IC_{50}$ to a value close to the MIC experimentally measured in the wild-type strain (Fig. 6B, black star; $MIC^{WT}$, ~125 $\mu$g/ml). These results show that the computational model of the bacitracin resistance network now precisely rebuilds the interplay of the two resistance modules and suggests that the simultaneous inductions of the two resistance modules jointly mediate lipid II cycle homeostasis under bacitracin treatment.

To further test the validity of our model, we demanded that it should be able to correctly predict the behavior of a $\Delta bcrC$ strain, in which BceAB is the sole genuine resistance determinant under bacitracin treatment (Fig. 6A, red data points). Due to the lack of BcrC in this strain, our model predicts that these cells produce higher levels of

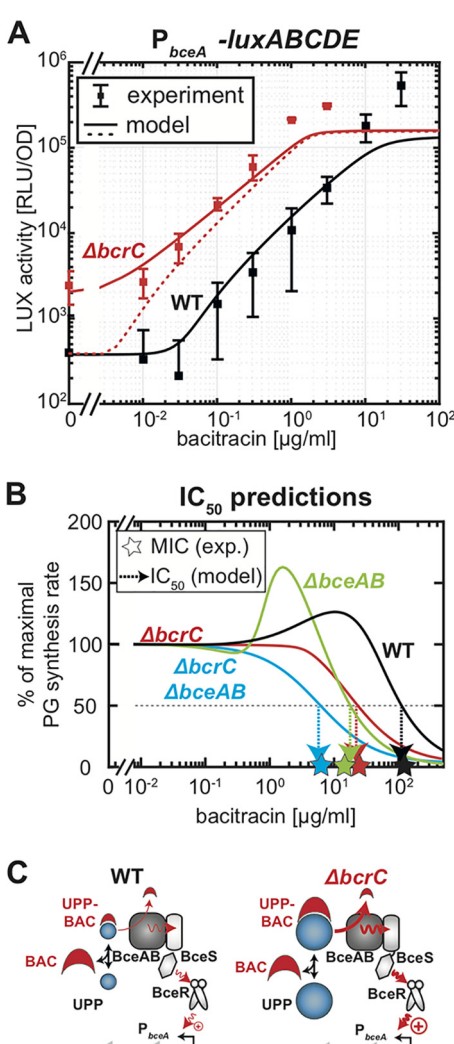

**FIG 6** Model calibration of the BceAB resistance module and MIC predictions in *B. subtilis* wild-type and mutant strains. (A) Fit of the full model for the bacitracin resistance network (including the BceAB module) to the experimental dose-response characteristic of the P$_{bceA}$-*luxABCDE* reporter in *B. subtilis* wild-type (WT) cells. The red dashed line shows the model prediction for the Δ*bcrC* mutant without invoking further fit parameters, revealing that although the model captures the overall increase in P$_{bceA}$ activity under bacitracin treatment, it does not describe the elevated basal promoter activity in the absence of bacitracin (see Text S1 for details). The red solid line shows the prediction of the model when the BceAB/BceS flux-sensing complex recognizes unbound UPP as a secondary substrate, which leads to a futile flux and triggers signaling in the BceRS two-component system. (B) The model behavior of the PG synthesis rate under bacitracin treatment for *B. subtilis* wild-type and mutant cells generates predictions for the respective IC$_{50}$ values (arrows), which are close to the experimental MIC values of the corresponding strains. (C) Schematic model behavior in the wild type and Δ*bcrC* mutant at identical bacitracin concentrations, illustrating that the higher UPP pool in the Δ*bcrC* mutant leads to higher UPP-BAC levels and thus to stronger activation of the BceRS signaling cascade than with the wild type.

LII intermediates (similar to the Δ*bcrC* Δ*bceAB* double mutant), which also lead to a higher abundance of UPP in the cell. This increased exposure of UPP then leads to higher levels of the bacitracin-bound form UPP-BAC in the presence of the antibiotic, which, in turn, serves as the stimulus for the activation of the P$_{bceA}$ promoter via the previously described flux-sensing mechanism. Accordingly, our model predicts that the accumulation of UPP-BAC in the Δ*bcrC* strain triggers an ~10-fold-higher P$_{bceA}$ promoter activity than with the wild-type strain (Fig. 6A, red dashed line), in qualitative agreement with the increased P$_{bceA}$ activity in the Δ*bcrC* strain determined experimentally (Fig. 6A, red data points). Also, without invoking any further parameter fitting, for a strain lacking BcrC the model-predicted IC$_{50}^{\Delta bcrC}$ of 22 μg/ml closely matches the

experimentally determined MIC of a Δ*bcrC* mutant (Fig. 6B; MIC$^{\Delta bcrC}$, ~25 μg/ml). This suggests that the simulated response of the BceAB resistance module—in conjunction with the elevated pool of LII intermediates—accurately capture the physiology of the system under bacitracin treatment.

One striking discrepancy between our model and the experimental data, however, was visible in the absence of bacitracin, where our experiments showed that the promoter activity of P$_{bceA}$ was also ~10-fold higher than in the wild type (Fig. 6A). This result is not compatible with the idea that UPP-BAC is the sole substrate for the flux-sensing mechanism via the BceAB transporter (triggering the activation of P$_{bceA}$), because UPP-BAC cannot be formed in the absence of bacitracin and thus the signaling mechanism should be inactive. However, it was previously hypothesized that UPP itself somehow triggers (futile?) ATP hydrolysis by BceAB and that high levels of UPP may contribute to the activation of P$_{bceA}$ (26, 36). Such an interaction seems plausible, given that the recognition of the UPP-BAC complex by BceAB likely involves interactions with both the UPP as well as the BAC moieties, raising the possibility that BceAB has some residual affinity for UPP. Interestingly, as noted above, the model predicts a significantly higher concentration of UPP in the Δ*bcrC* mutant than in the wild type (Fig. 5C), as caused by the expected increase of the overall concentration lipid II cycle intermediates as well as the reduced overall phosphatase activity in the absence of BcrC. To test if futile activation of P$_{bceA}$ by these elevated UPP levels can explain the higher basal promoter activity observed in the Δ*bcrC* mutant, we modified the theoretical description of the total load per BceAB transporter, $J_{load}$, which is proportional to the rate of ATP hydrolysis and, in turn, regulates the promoter activity of P$_{bceA}$, as follows:

$$J_{load} = J'_{BAC} + J_{futile}$$

where

$$J'_{BAC} = \frac{[\text{UPP-BAC}]}{K_m} \Big/ \left(1 + \frac{[\text{UPP-BAC}]}{K_m} + \frac{[\text{UPP}]}{\widetilde{K}_m}\right)$$

describes the flux of bacitracin released from UPP-BAC complexes and

$$J_{futile} = \frac{[\text{UPP}]}{\widetilde{K}_m} \Big/ \left(1 + \frac{[\text{UPP-BAC}]}{K_m} + \frac{[\text{UPP}]}{\widetilde{K}_m}\right)$$

describes the rate of futile ATP hydrolysis triggered by UPP alone, with Michaelis constants $\widetilde{K}_m$ and $K_m$ describing the binding constants of the transporter for UPP and UPP-BAC, respectively. Within these equations an increasing level of UPP increases the overall load per transporter, $J_{load}$, which then triggers signaling and activation of P$_{bceA}$, while the same increase in UPP leads to a reduction of the flux of bacitracin release, $J'_{BAC}$, induced by competitive binding to the transporter. Strikingly, by modifying the model as depicted, the predictions of P$_{bceA}$ promoter activation differed significantly between the simulated scenarios (Fig. 6A, red solid line). While the model output for the wild-type equaled the prediction of the former model, the modified model predicted a significant elevation of P$_{bceA}$ promoter activities in a strain lacking BcrC, which closely resembles the experimental data for the Δ*bcrC* strain (Fig. 6A). Thus, our results of the modified model support the hypothesis of futile activation of P$_{bceA}$ by UPP under BcrC deletion (Fig. 6C). However, since the model modifications did not affect the IC$_{50}$ predictions of the various mutants (see Text S1), the effect of UPP on P$_{bceA}$ activation is likely negligible under bacitracin treatment and solely affects the level of basal promoter activity.

## DISCUSSION

Building on the experimental characterization of the bacitracin resistance network in *B. subtilis* (26), we here present the first theoretical description of this regulatory network. The mathematical model developed in this study not only accurately predicts the efficacy of bacitracin in the wild type and various mutant strains lacking one or both of the resistance modules (Fig. 6B) but also uncovers important regulatory features of

mSystems®

the resistance network. By successively incorporating mathematical descriptions of the individual resistance modules into a preexisting theory of the lipid II cycle, we showed that the interplay between the two major resistance determinants (BceAB and BcrC) is strictly linked to the properties of the lipid II cycle, which change in response to bacitracin.

One important insight of our analysis is that BcrC is the more dominant UPP phosphatase than UppP, dictating the bulk of the overall UPP recycling rate in the lipid II cycle of *B. subtilis*. This is also reflected by the fact that *bcrC* expression is significantly elevated under bacitracin stress (Fig. 2Bi), while *uppP* is constitutively expressed (14–16), implying even more pronounced changes in the total phosphatase activity in response to bacitracin than previously appreciated. These results are in accordance with experiments showing that a *bcrC* deletion significantly reduced the resistance toward bacitracin in *B. subtilis*, while a deletion of *uppP* had only moderate effects (14, 16). In fact, to ensure a strong protective effect in response to cell envelope stress, it seems physiologically plausible for the cell to activate expression of the phosphatase contributing most strongly to the progression of the lipid II cycle.

Another finding arising from the combination of theory and experiment was the homeostatic control of lipid II cycle intermediate levels in a Δ*bcrC* mutant, ensuring the close-to-optimal progression of the cycle despite the lack of the important phosphatase BcrC. We found two σ$^M$-controlled genes, *ispD* and *ispF*, involved in the *de novo* synthesis of UPP, to be significantly upregulated, which counteracted the depletion of the lipid II pool caused by a shortage of UPP phosphatase activity. While we did not directly prove that this leads to an increase in the overall abundance of lipid II cycle intermediates, our experimental and theoretical results indirectly support this hypothesis in three ways. (i) The Δ*bcrC* Δ*bceAB* double mutant is significantly more resistant to bacitracin than naively predicted by a model with constant total lipid II cycle intermediate pools, suggesting that a compensatory upregulation of these pools contributes to bacitracin resistance in this mutant. (ii) Under bacitracin treatment, the P$_{bceA}$ promoter is ~10-fold more active in a Δ*bcrC* mutant than in the wild type, suggesting that the major substrate, UPP-BAC, of the BceAB transporter is more abundant in the mutant, which, in turn, triggers stronger activation of P$_{bceA}$ via the flux-sensing mechanism described in reference 21. (iii) Even in the absence of bacitracin, the Δ*bcrC* mutant displays an ~10-fold-higher P$_{bceA}$ activity than the wild type, suggesting that the elevated UPP pool in this mutant is sufficient to trigger some futile ATP hydrolysis by the BceAB transporter, which then activates P$_{bceA}$ via the flux-sensing mechanism.

From a systems-level perspective, the upregulation of lipid carrier production seems to be a particularly elegant way to maintain cycle homeostasis under antibiotic treatment, because it naturally preserves the relative balance between the different lipid II cycle intermediates. In fact, in a closed-loop system like the lipid II cycle, the stoichiometry between the intermediate pools is determined only by the catalytic rates and abundances of the enzymes catalyzing cycle progression and not the overall abundance of all intermediates (34). Thus, the sequestration of one cycle intermediate by an antibiotic (such as bacitracin, vancomycin, or nisin) will lead to the stoichiometric reduction of all other intermediates. One possible way to accelerate lipid II cycle progression would be the simultaneous upregulation of all lipid II cycle-associated enzymes. In contrast to such a fine-tuned, orchestrated regulation, our results suggest that the cell compensates this shortage by *de novo* synthesis of cycle intermediates, which rapidly equilibrates among the different stages of the lipid II cycle intermediates and naturally replenishes intermediate levels in the correct stoichiometry. We suggest that this strategy implements a robust way of ensuring lipid II cycle homeostasis.

Although we did not decipher the exact stimulus for activation of the BcrC resistance module, our theory revealed that the regulation of *bcrC* expression and regulation of *bceAB* expression are tightly interconnected via the properties of the lipid II cycle itself. Since the activation of the resistance determinants in response to bacitracin go along with significant changes in the concentrations of the different lipid II cycle intermediates, it is plausible that not only BceAB but also the BcrC resistance module

mSystems®

somehow responds to these changes. Indeed, it seems advantageous to regulate the overall resistance against bacitracin by responding to changes in the properties of the lipid II cycle, since this does not demand additional regulatory structures for each resistance module, which might be costly to produce and would further complicate the resistance network. More generally, monitoring the physiological state of the pathway itself may serve as a cost-effective strategy to regulate the interplay between the different resistance determinants protecting the cell against cell envelope stress.

Ultimately, this study clearly highlights how mathematical modeling provides a better understanding of sophisticated cellular responses toward environmental conditions, in particular antibiotic treatment. By combining existing theoretical descriptions of the various modules of the cellular response, a comprehensive model of the complex network structure evolved. Successive integration of additional modules of the cellular response into the growing model enabled us to study both the basal regulatory features of every individual layer and the factors determining the interplay between them within the whole network. We showed that a simple existing model can be expanded to develop a more and more complex picture, and eventually the model itself could even become a building block when describing a network on a broader scale. This approach can act as a blueprint for acquiring true systems-level understanding of complex regulatory structures, describing not only the organization of resistance system against other antibiotics but also more generally multitiered response networks that can be expected across many bacterial species and a range of environmental stressors.

## MATERIALS AND METHODS

**Bacterial strains and growth conditions.** *Bacillus subtilis* and *Escherichia coli* were routinely grown in lysogeny broth (LB medium) at 37°C with agitation (200 rpm). Transformations of *B. subtilis* were carried out as described previously (37). All strains used in this study are derivatives of the wild-type strain W168 and are listed in Table S1. Kanamycin (10 mg ml$^{-1}$), chloramphenicol (5 mg ml$^{-1}$), tetracycline (10 mg ml$^{-1}$), and erythromycin (1 mg ml$^{-1}$) plus lincomycin (25 mg ml$^{-1}$) for macrolide-lincosamide-streptogramin B (MLS) resistance were used for the selection of the *B. subtilis* mutants used in this study. Solid media contained 1.5% (wt/vol) agar.

**Luciferase assays.** Luciferase activities of *B. subtilis* strains harboring pAH328 derivatives were assayed using a Synergy2 multimode microplate reader from BioTek (Winooski, VT), essentially as described in reference 26. Briefly, the reader was controlled using the software Gen5 (version 2.06). Cells were inoculated 1:1,000 from fresh overnight cultures and grown to an optical density at 600 nm (OD$_{600}$) of 0.1 to 0.5. Subsequently, cultures were diluted to an OD$_{600}$ of 0.01 and split into 100 $\mu$l per well in 96-well plates (black walls, clear bottom; Greiner Bio-One, Frickenhausen, Germany). Cultures were incubated at 37°C with linear agitation (medium intensity), and the OD$_{600}$ and luminescence were monitored every 3 min. After 1 h, freshly diluted Zn$^{2+}$-bacitracin was added to the desired final concentrations, and incubation and monitoring every 3 min were resumed for 8 h. Specific luminescence activity is given by the raw luminescence output (relative luminescence units [RLU]) normalized by cell density (RLU/OD$_{600}$). Please note that the luminescence sensitivity of the microplate reader used in this study is lower than in our previous work (26), such that for a given strain the relative luminescence units reported here are ~10- to 50-fold lower than previously reported. Also, in the present work we used a more stringent cleaning procedure for test tubes (used for overnight and day cultures), which completely avoids the use of detergents and instead relies on mechanical cleaning and autoclaving only. With this procedure we found that the basal expression of cell wall stress modules (in the absence of antibiotics) is slightly lower than with the previous procedure, explaining, for instance, why the observed luminescence activity of the P$_{bceA}$-*lux* construct in Fig. 2Biii remained at a low level between 0 $\mu$g/ml and $3 \times 10^{-2}$ $\mu$g/ml of bacitracin and increased only at higher antibiotic concentrations, while we found a slight increase of luminescence activity at these concentrations before (26). These subtle differences in expression levels affected the qualitative behavior only under noninduced conditions (no bacitracin) and neither impacted the development of the computational model nor affected the conclusions from this work.

**MIC assays.** For concentration-dependent growth experiments, cells were grown as described for the luciferase assays and OD$_{600}$ was measured analogously. The growth rate within the first hour after bacitracin addition was determined to monitor the concentration-dependent effects of bacitracin on cell growth (Fig. 2Bi). The MIC was defined as the concentration of antibiotic that fully inhibited growth, i.e., for which the growth rate equals zero.

**Relative quantitative RT-PCR.** *Bacillus subtilis* W168 and Δ*bcrC* cells were collected at an OD$_{600}$ between 0.3 and 0.5 (measured in a spectrophotometer) and suspended in TRIzol (Ambion). The cells were lysed through bead beating with 0.1-mm zirconia beads. RNA was extracted from exponentially growing cells with TRIzol reagent. DNA was removed with DNase (Thermo Scientific) and the DNase was then heat deactivated in the presence of EDTA. RT-qPCR was performed with the Luna universal one-step

RT-qPCR kit (New England Biolabs). One microliter of 10-fold-diluted RNA was added to 4 $\mu$l of RT-PCR mix and subjected to a reverse transcription step at 55°C and 45 cycles of PCR (10 s at 95°C and 30 s at 60°C). The average threshold cycle ($C_T$) value of three technical replicates of three biological replicates for each sample was used in $\Delta\Delta C_T$ relative expression analysis (38). The reference genes were the constitutively expressed genes *recA* (BSU16940) and *gyrB* (BSU00060).

**Computational model and simulations.** A detailed description of the model assumptions and equations for the bacitracin resistance network and additional analyses of the model are given in Text S1. The numerical calculations of the differential equations of the model as well as the individual simulations were performed with custom scripts developed in MATLAB software (The MathWorks, Inc.).

## SUPPLEMENTAL MATERIAL

Supplemental material is available online only.

**TEXT S1**, DOCX file, 0.04 MB.
**FIG S1**, PDF file, 2.6 MB.
**FIG S2**, PDF file, 0.9 MB.
**TABLE S1**, DOCX file, 0.01 MB.
**TABLE S2**, DOCX file, 0.01 MB.
**TABLE S3**, DOCX file, 0.02 MB.

## ACKNOWLEDGMENTS

This work was supported by the LOEWE Program of the State of Hesse (SYNMIKRO support to G.F.), the Deutsche Forschungsgemeinschaft (DFG grants FR3673/1-2 to G.F. and MA2837/2-2 to T.M. in the framework of DFG priority program SPP1617) and the Biotechnology and Biological Sciences Research Council (BBSRC grant BB/M029255/1 to S.G.). H.P. was supported by the Cusanuswerk scholarship program (Germany), and C.M.K. was supported by a University of Bath Research Studentship Award.

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
