## [Reviewer comments · mSystems]

From modules to networks: A systems-level analysis of the bacitracin stress response in *Bacillus subtilis*

Hannah Piepenbreier, Andre Sim, Carolin Kobras, Jara Radeck, Thorsten Mascher, Susanne Gebhard, and Georg Fritz

Corresponding Author(s): Georg Fritz, The University of Western Australia

Review Timeline:

Submission Date:	October 30, 2019
Editorial Decision:	December 19, 2019
Revision Received:	December 20, 2019
Accepted:	December 28, 2019

Editor: Domitilla Del Vecchio

Reviewer(s): The following individuals involved in review of your submission have agreed to reveal their identity: Rushina Shah (Reviewer #2)

Transaction Report:

DOI: <https://doi.org/10.1128/mSystems.00687-19>

Reviewer #1 (Comments for the Author):

To understand the mechanisms of bacterial resistance against antibiotics, authors set up a Michaelis-Menten-kinetics-based model of the lipid II cycle of *B. subtilis* in order to rationalize their experimental data in response to bacitracin treatment. Specifically, the model integrated a BecAB module and a BcrC module. The model was justified by its predictive power on the minimal inhibitory concentration (MIC) of bacitracin causing zero growth rate in *B. subtilis* wild-type strain W168, a $\Delta bceAB$ mutant, a $\Delta bcrC$ mutant and a $\Delta bceAB \Delta bcrC$ double mutant. The authors claimed that the full model uncover important regulatory features of the resistance network. The features included a high level of cross-regulation between the two major resistance modules and the lipid II cycle of cell wall biosynthesis.

To evaluate the research correctly, I exam the model and the experiments used in this study as follows.

The model of the lipid II cycle (Piepenbreier et al., 2019) was submitted for the review. It is unclear whether this model is justified and accepted by the reviewers or not. The IC₅₀ value was defined as the antibiotic concentration reducing the PG synthesis rate to 50 % of its unperturbed rate. The PG synthesis rate is only estimated from the model, not from the experimental data. Is it possible to measure the PG synthesis rate as a direct evidence to verify the model of the lipid II cycle? The direct measurement of PG synthesis rate is also important to verify the present integrated model which predicts the experimentally determinable MIC value by the IC₅₀ value.

As the reviewer notes, this work is built on a previous mathematical model of the lipid II cycle, which was accepted for publication in *Nature Communications* a few days before the present submission (Piepenbreier et al., 2019; see <https://doi.org/10.1038/s41467-019-10673-4>). Given that three expert reviewers found our analysis to be sound, and that the work was picked as editor's highlight (<https://www.nature.com/collections/xqqjnjxkct/content/sonja-schmid>), we feel safe to say that the model is considered to be justified.

As the reviewer suggests, we would also love to get our hands on an experimental measurement of the PG synthesis rate, but to date such measurements are highly nontrivial and far beyond the scope of this work. In fact, obtaining quantitative measurements of the lipid II cycle intermediates itself requires a set of extensive biochemical experiments only a few labs in the world are capable of. Independent of experimental measurements, however, our previous work showed that if one considers a 50% inhibition of the PG synthesis rate (relative to the unperturbed wild-type rate) as being lethal, the theoretical model was able to consistently predict the experimental MICs of 5 cell wall antibiotics targeting different intermediates of the lipid II cycle (Piepenbreier et al., 2019). This indicates that the IC₅₀ of PG biosynthesis is a good theoretical predictor for the experimental MICs.

To evaluate how changes in this assumption would affect our current analysis, we considered the hypothetical scenario in which another level of PG inhibition, i.e. 30% (IC₃₀), is lethal to cells. As shown in Figure R1, this will lead to an increase in the predicted MIC values for wildtype and all mutant strains, in disagreement with the experimental MICs

(indicated as stars in the figure). Thus, although the PG synthesis rate is not experimentally accessible to us, we think that the alignment between theoretically predicted IC_{50} values and experimental MICs for different strains (this work) and antibiotics (Piepenbreier et al., 2019) can be considered as a verification for our assumption that the IC_{50} is a solid predictor for the MIC.

Figure R1. Experimental MIC values for wildtype and indicated mutant strains (*stars along the x-axis*) are closely resembled by the theoretical prediction of the IC_{50} values (*solid arrows*), while the IC_{30} values lead to an ~2x overestimation of the experimental MICs for all strains (*dashed arrows*).

The authors reproduced most data in Figure 2B (i) and 2B (iii) which were published in the previous work (Radeck et al., 2016) except the new data using TMB1632 strain. However, there were significant differences between the current results and the previous data. For instance, the black and the green curves in Figure 2B (i) should have a similar Lux activity in the absence of bacitracin. Comparing to the previous data (Fig.3A of Radeck et al., 2016), the shape of the black curve in Figure 2B (iii) was not monotonically increasing and the standard deviations were much larger. Importantly, this black curve in Figure 2B (iii) was used for model calibration in Figure 5A. The discrepancy to the previous data raises the question whether the full model is truly able to capture the responses to bacitracin and whether the model's predictions are correct. Clearly, the black solid line fitting by the model in Figure 5A cannot capture the results of the same P_{bceA} -lux construct in the same strain in the same conditions as the black curve shown in the Fig. 3A of the previous work (Radeck et al., 2016). Authors need to reproduce the previous data correctly and provide the expression dynamics of resistance modules in all mutant backgrounds in the Supplementary Information.

The reviewer is right that there are a few quantitative differences between the data reported here and our previous data published in (Radeck et al., 2016). One of the underlying reasons is that in the current study we had to rely on a different plate reader, which has a lower luminescence sensitivity than the one used in the previous study. Based on this, the relative luminescence units (RLU) reported here are ~10-50 fold lower than in the previous study. A second aspect is that our experimental conditions have changed slightly since our paper in 2016: We now use a more stringent cleaning procedure for our test tubes, which completely avoids the use of detergents and instead relies on mechanical cleaning + autoclaving only. With this procedure we found that the basal expression of cell wall stress modules (in the absence of antibiotics) is slightly lower than with the previous procedure, explaining, for instance, why the observed luminescence activity of the P_{bceA} -lux construct in Figure 2B (iii) remains at a low level between 0 $\mu\text{g/ml}$ and 3×10^{-2} $\mu\text{g/ml}$ bacitracin and only increases at higher antibiotic concentrations. Thirdly, at low bacitracin levels the somewhat higher standard deviations between replicates (compared to our previous study) emerge from the fact that we operate closer to the detection limit of the plate reader. Given these standard deviations, the slight decrease in luciferase activity from 0 $\mu\text{g/ml}$ to 3×10^{-2} $\mu\text{g/ml}$ bacitracin is not significant. We hope the reviewer will concur that these experimental data are in fact compatible (within error bars) with our model of monotonically increasing promoter activities, as shown in Figure 5A.

The reviewer also noticed that in the present study the P_{bcrC} -lux construct showed a ~2-fold lower activity in a $bceAB$ mutant compared to the wild-type (green vs. black curve in Fig. 2B (i)), while in our previous study both strains showed similar basal P_{bcrC} -lux activities. This small difference could again be the result of the changes in our experimental procedures, where trace amounts of detergents in our previous data might have triggered elevated P_{bcrC} activity in the $bceA$ mutant. However, given the size of the error bar on the P_{bcrC} activity in the wild-type, it seemed conservative to assume that there is no significant change in the basal $bcrC$ expression levels between $bceAB$ mutant and wildtype, as reflected in our computational model described in the Supplementary Text.

To address the subtle differences between the data in this and our previous work, we now added the following paragraph to the description of the Luciferase assays in the Methods section (lines 172-186):

“Please note that the luminescence sensitivity of the microplate reader used in this study is lower than in our previous work (Radeck *et al.*, 2016), such that for a given strain the relative luminescence units (RLU) reported here are ~10-50 fold lower than previously reported. Also, in the present work we used a more stringent cleaning procedure for test tubes (used for overnight- and daycultures), which completely avoids the use of detergents and instead relies on mechanical cleaning and autoclaving only. With this procedure we found that the basal expression of cell wall stress modules (in the absence of antibiotics) is slightly lower than with the previous procedure, explaining, for instance, why the observed luminescence activity of the P_{bceA} -*lux* construct in Figure 2B (iii) remains at a low level between 0 $\mu\text{g/ml}$ and 3×10^{-2} $\mu\text{g/ml}$ bacitracin and only increases at higher antibiotic concentrations, while we found a slight increase of luminescence activity at these concentrations before (Radeck *et al.*, 2016). These subtle differences in expression levels only affect the qualitative behavior under non-induced conditions (no bacitracin), and neither impact the development of the computational model, nor affect the conclusions from this work.”

The Figure 2B (ii) showed the impact of bacitracin on growth. It is unclear the impact was analyzed for the BcrC module or the BceAB module. The supplemental Figure S1 provided the growth curves for all strains used in this study. What were the data used to generate the Figure 2B(ii)? Importantly, which module’s MIC values were determined in the Figure 2B(ii) in order to verify the prediction power of the model? In addition, since the MIC values are important to verify the model predictions, authors need to describe how they estimate the MIC values for the respective strain. For instance, none of the reported values such as ~6.3 $\mu\text{g/mL}$ on Line 239, ~14.5 $\mu\text{g/mL}$ on Line 240, ~125 $\mu\text{g/mL}$ on Line 398, and ~25 $\mu\text{g/mL}$ on Line 415 was the concentration of bacitracin directly used in the experiments (Supplementary Fig. S1). Especially, ~125 $\mu\text{g/mL}$ on Line 398 should be a deduced value, not a true experimental value because the maximal concentration of bacitracin was only 100 $\mu\text{g/mL}$.

The reviewer is absolutely right that we missed to specify the calculation of growth rates in Figure 2B(ii). Given that there was no discernible impact of the reporter construct (P_{bcrC} -*lux* or P_{bceA} -*lux*) on cell growth (see Supplementary Figure S1), we simply averaged the growth rate for a given strain transformed with the different reporter constructs.

In line with this, the model does not distinguish between the different reporter strains and predicts the MICs for the wild-type and the different mutants independent of the reporter constructs. To better explain this, we amended the legend to Figure 2B (ii):

“Panel (ii) shows the corresponding normalized growth rates, which represent the averaged growth rates of strains carrying the different reporter constructs, but sharing the same genotype.”

As a common strategy in MIC assays, we defined the MIC as the antibiotic concentration that fully inhibits cellular growth, that is, the antibiotic concentration at which the growth rate is zero for the first time. This concentration was determined by a linear interpolation between the experimental data points. The exact MIC is then the zero crossing of the linear function. To determine the MIC for the wild-type, the linear function between the last two data points was extrapolated. To clarify this calculation of the MIC, we amended the legend of Figure 2B (ii):

“The coloured stars indicate the experimental MIC values, calculated as the concentration at which a linear interpolation between the data points crosses the zero line. In case of the wild-type strain, the MIC was calculated by a linear extrapolation to the zero line, given that the highest concentration tested did not fully inhibit growth.”

According to the caption of the Supplemental Figure S1, the exponential growth phase and its rate were defined and determined within the first hour after bacitracin induction. Referring to the Supplemental Figure S1, within the time window (60-120 min), OD₆₀₀ values are below 0.3. However, in Line 184, the samples for RT-qPCR were collected at OD₆₀₀ between 0.3-0.5, where is clearly the region out of exponential phase because the slope of the growth curves changes. But, in Line 186, authors reported that RNA was extracted from exponentially growing cells for the RT-qPCR. In Line 316, authors claimed that during exponential growth, expression of both *ispD* and *ispF* was 2-fold higher in the *bcrC* deletion strain compared to the wild-type strain. Since the samples were clearly not collected from exponential growth, the conclusions of RT-qPCR experiment are inaccurate.

We thank the reviewer for pointing out this apparent glitch, which simply emerges from the fact that the OD₆₀₀ readings in the plate reader are not directly comparable to a “standard” OD₆₀₀ reading in a spectrophotometer. In fact, to determine the OD of samples for RT-qPCR we performed standard OD readings with cell cultures filled into cuvettes with a standard 1 cm light path. In contrast, our plate reader measures absorption of a light beam crossing the microplate well from top to bottom, such that the light path within the culture depends on the filling height of the well. Additionally, standard OD measurements are to be performed in the linear detection range of the spectrophotometer, which is achieved by diluting samples to the appropriate OD. Altogether, these factors lead to a proportionality factor of 7.4 between the OD₆₀₀ measured in our plate reader and the OD₆₀₀ measured with a spectrophotometer (see Figure R2). Accordingly, when inspecting the growth curves in Supplemental Figure S1 (measured in our plate reader), it becomes apparent that the exponential growth phase extends until an OD₆₀₀^{plate reader} of 0.2, corresponding to an OD₆₀₀^{spectrophotometer} of ~1.5. Therefore, collecting the samples for RT-qPCR between an OD₆₀₀^{spectrophotometer} of 0.3 and 0.5 is clearly within the exponential growth phase, as claimed by us. However, we totally agree that this was not clear in the previous version of the manuscript and now clarify this point by highlighting the difference between the two OD readings both in the Methods section as well as in the legend to Figure S1:

“Please note that the numerical values of the optical density at 600 nm (OD) detected by this microtiter plate reader are ~7.4-fold lower than the values of an OD₆₀₀ measurement in a spectrophotometer, using a cuvette with a standardized 1 cm path length in the bacterial culture.”

Figure R2. Standard curve for the optical density (OD₆₀₀) of *B. subtilis* cultures measured in a spectrophotometer (x-axis) and measured in a plate reader (y-axis). The trendline shows a linear fit with zero y-axis intercept, and from the slope of the line we infer a ratio of $OD_{600}^{\text{spectrophotometer}}/OD_{600}^{\text{plate reader}}=7.4$.

In addition, in Line 189, authors only used 10-fold diluted RNA for RT-qPCR measurement. Authors need to show at least three different dilutions of RNA samples for the RT-qPCR measurement to ensure the measurement in the linear detection range to increase the confident level of the RT-qPCR data. Also, in Line 194, authors need to provide supporting

evidences or references to justify the use of *recA* and *gyrB* as good reference genes in the present experimental conditions.

We totally agree with the reviewer and we of course performed these routine controls. In fact, in Figure R3 we show for our two reference genes that the 10-fold dilution of RNA samples fell in the linear detection range for qPCR. Accordingly, we now state in the Methods part that our 10-fold dilution falls into the linear detection range.

Figure R3. Standard curve for different dilution factors of RNA samples (x-axis) against the cycle threshold (CT), defined as the number of cycles required for the accumulating fluorescent signal to cross a pre-defined threshold.

In qPCR experiments *gyrB* and *recA* are well known reference genes for bacteria, given their invariable expression across many experimental conditions (da Silva et al. 2016; Crawford et al. 2014; Gomes et al. 2018; Reiter, Kolstø, and Piehler 2011). Both genes have roles independent of *bcrC* and therefore should be unaffected by its deletion. Apart from those two genes, we also considered *rpsU* and *mnaA* as additional, well-known reference genes for *Bacillus* species (Reiter, Kolstø, and Piehler 2011). However, we found that *rpsU* was too highly expressed in comparison to our other test genes, making normalization unreliable. Similarly *mnaA*, encoding an UDP-N-acetylglucosamine 2-epimerase, is involved in teichoic acid synthesis and thus closely linked to cell wall biosynthesis. Given that our preliminary experiments showed that *mnaA* was differentially expressed between wild-type and a *bcrC* deletion strain, we also discarded *mnaA* as a reference gene.

As a proof of concept that *gyrB* and *recA* are reasonable reference genes for RT-qPCR for the strains used here, we found that by applying this normalization to the expression levels of *uppS* and *uppP*, the resulting values were invariant between the wild-type and the *bcrC* mutant (Figure 4B in the main text and Figure R4). This is consistent with the fact that both genes are dependent on the housekeeping sigma factor σ^A , while the *bcrC* mutant triggers upregulation of the σ^M regulon (Eiamphungporn and Helmann 2008). Based on these considerations we hope the reviewer agrees that *gyrB* and *recA* are suitable reference genes that allow interpretation of the qPCR data in our strains.

It is unclear whether the term xBcrC, which describes the bacitracin-dependent contribution from BcrC in line 258, includes the description of the basal high expression level at most culture conditions (Radeck et al., 2017b). In Line 344, BcrC was predicted as the dominant phosphatase (xBcrC=63 %). By reading this, it seems imply that the term xBcrC includes both the bacitracin-dependent contribution and the basal high expression level. Could the author clarify what the term xBcrC actually describes?

We thank the reviewer for commenting on this point, which also lead to some confusion for reviewer #2. To answer the question, x^{BcrC} reflects the fraction of the total phosphatase activity incurred by BcrC in the absence of bacitracin, i.e. at the basal expression level of *bcrC*. Accordingly, a value of $x^{BcrC}=63\%$ indicates that without bacitracin stress BcrC is the dominant phosphatase.

To clarify the meaning of the term x^{BcrC} in the manuscript, we improved the visual representation in Figure 3 and expanded the description in the main text (lines 274-285) and the legend of Figure 3 (738-741).

“Thus, the total speed of the UPP dephosphorylation reaction is proportional to the weighted sum of the bacitracin-dependent contribution from BcrC and the bacitracin-independent contribution from UppP (Fig. 3C), as indicated in Eq. (1).

$$(1) \quad \text{rate of UPP dephosphorylation} \sim x^{BcrC} \times f^{BcrC}(BAC) + [1 - x^{BcrC}] \times 1 .$$

Here, the factor x^{BcrC} quantifies the fractional contribution of BcrC and $1 - x^{BcrC}$ the fractional contribution of UppP to the total phosphatase activity in the absence of bacitracin, respectively (x^{BcrC} takes values between 0 and 1). Moreover, the upregulation of BcrC levels under bacitracin treatment leads to a fold-induction of the UPP phosphatase activity according to a factor $f^{BcrC}(BAC)$, which ranges from 1 to a maximal fold-change (Fig. 3A). Thus, the stronger the contribution of BcrC towards the overall phosphatase activity (higher x^{BcrC}), the more pronounced the acceleration of the UPP dephosphorylation reaction in response to bacitracin (Fig. 3C).”

Minor points about the presentation in writing:

- The authors need to clearly indicate whether the data are the new data in the present work or the reproduced data of the previous work.
 - o For example, in Line 240, ... compared to the $\Delta bceAB$ mutant, “consistent with” earlier results. It would be more appropriately to say “reproducing” earlier results.

We would like to stress that none of the plots, nor the underlying data, shown in this work is a simple “reproduction” of previous work. Albeit some of the strains used here were characterized before, the central comparison to the $\Delta bcrC\Delta bceAB$ strain is genuinely new and required the re-characterization of all strains under the experimental conditions used in this study. Thus, we feel that our statements that the new results are consistent with earlier results is most accurate.

- The full name of MIC was described in line 237 after the MIC first showed on line 140.

Corrected.

- Figure 2A is the key figure about the novelty of this work which integrated two modules in the lipid II cycle. However, it was only mentioned very briefly in the introduction and was not described at all in the result but in the caption of Figure 2A.

We thank the reviewer for pointing this out. We now explain Figure 2A in more detail in the first section of the Results part, where the structure of our work is laid out in some detail (lines 221-231).

- The results between Line278 and 329 were frequently referred to Supplementary Text and Supplementary Fig S2. Should the authors consider to move these results to the main figure?

That is a great idea – we moved Supplementary Fig S2 into the main text (new Fig. 5), which makes it much easier for the reader to follow the argument.

- The order to describe the results in Figure 5 is 5A, 5C, and finally 5B. Could the authors find a better way to describe the results in an order like A, B, and then C?

We swapped panels B and C in the Figure (now Figure 6), adhering to the narrative of the text.

- Line 142, an increased total number of lipid carriers. Are lipid carriers UP or UP and UPP?

The total number of lipid carriers are UP + UPP + lipid I + lipid II. We now specify this in the text (lines 140-141).

- Line 167, it should be 100 uL, not 100 mL.

Thanks – corrected!

- The order of the presentation in writing is not in the order of numbering of the figures.

We double-checked the numbering of the figures and made sure that the figure numbers adhere to their first appearance in the text. However, the reviewer is right that in some cases we do not adhere to the order when it comes to referencing the subpanels A, B, C, etc. of figures, because we feel that it is more intuitive to group logically similar panels together in a figure than strictly obeying sequential referencing when it comes to subpanels. As we find this to be common practice (see for instance a recent mSystems article here <https://msystems.asm.org/content/4/5/e00416-19>), we would like to ask for permission to keep referencing to subpanels as is.

- Line 255, UppP should be introduced in lipid II cycle in the introduction, not just in the caption of Figure 1. BcrC and UppP and YodM should be mentioned in the introduction to let readers know that there are three UPP phosphatases in *B. subtilis* and their relative importance.

We inserted a brief description of the lipid II cycle-associated enzymes in the introduction, explicitly mentioning BcrC, UppP and YodM:

“Briefly, within this essential pathway the peptidoglycan (PG) precursors N-Acetylglucosamine (GlcNAc) and N-Acetylmuramic acid (MurNAc)-pentapeptide are sequentially attached to the lipid carrier molecule undecaprenyl phosphate (UP) by MraY and MurG, thereby forming lipid II (Fig. 1). Subsequently, lipid II is flipped across the cytoplasmic membrane via the flippases MurJ and Amj, where the PG monomer (GlcNAc-MurNAc-pentapeptide) is incorporated into the growing cell wall by various redundant penicillin-binding proteins (PBPs). This leaves the lipid carrier in its pyrophosphate form (UPP), which has to be recycled to UP by dephosphorylation (in *B. subtilis* via the UPP phosphatases BcrC, UppP and, to a minor degree, YodM (Cao and Helmann, 2002; Zhao *et al.*, 2016; Radeck *et al.*, 2017b)) to allow a new round of PG monomer transport.”

- Comparing the legends in Figure 2B(i), 2B(iii) and Figure 5A to the genotype in the Supplementary Table 1 and 2, should it be PbecA-lux, or PbecAB-lux? Which promoter was used to drive the expression of lux reporter?

Thanks for pointing this glitch out! We now corrected all nomenclature in the text and figures to PbceA-lux, which is the most accurate way of referring to the promoter in front of *bceA*.

- Which strain does the WT legend mean in Figure 2 B(ii)? Dose WT mean TMB1619 or TMB1620 strain?

We now clarified this with the following addition to the figure legend:

“Panel (ii) shows the corresponding normalized growth rates, which are obtained as the average growth rates of strains carrying the different reporter constructs, but sharing the same genotype.”

- In Line 331, it is unclear which growth data readers should compare to Supplementary Figure S1.

We now specified the statement more clearly:

“... as suggested by the fact that growth of the *bcrC* deletion strain was not drastically reduced when compared to the wildtype.”

- In Line 390, about the Supplement Table 3, for the BecAB module, there is no Fig. 2C in the main text about the parameter of fold-change of PbceAB promoter.

Corrected to refer to Fig. 2B(iii).

- Line 432, there is no Supplementary Figure 3B.

We corrected reference to this figure, which has now become Fig. 5C.

- Line 702-704, the description of the used strains should be improved for clarity. For instance, there is no PbecA-lux in the $\Delta bceAB\Delta bcrC$ strain. The writing is misleading.

To be clearer, we now explicitly state the genotypes of the strains in the figure legend.

Reviewer #2 (Comments for the Author):

Review mSystems: From modules to networks:

A systems-level analysis of the bacitracin stress response in *Bacillus subtilis*

This paper analyzes two mechanisms with which the bacteria *B. subtilis* achieves drug-resistance against a cell-wall synthesis inhibiting antibiotic bacitracin. The work analyzes this problem experimentally, as well as through a computational model. Experimentally, the effect of the antibiotic is observed on 4 types of strains: the wild type (WT), cells without BceAB, cells without BcrC, and cells without BceAB and BcrC. The computational model consists of two previous models (one that describes the lipid II cycle and one that describes action of BceAB), and further adds to this model a third module - the BcrC module. The work is novel and interesting in that it analyzes the interplay of these two distinct mechanisms of drug resistance in the bacteria, both via experiments as well as a computational model.

Questions that arise:

1- Why is the UppP activity scaled by $(1-x_{\text{BcrC}})$? Is this term to reflect the competition between the two phosphatases, so that if one is increased, the other decreases? Please explain this scaling, since it appears that in the model, knocking out BcrC implies setting $x_{\text{BcrC}} = 0$. Do we expect the phosphatase activity of UppP to go up in this strain? Is this experimentally validated if so?

We thank the reviewer for commenting on this point, which was also mentioned by reviewer #1. As explained above, x^{BcrC} quantifies the fraction of the total phosphatase activity incurred by BcrC in the absence of bacitracin. In total, the unaffected phosphatase activity (without bacitracin), which originates from UppP and BcrC together, is assumed to be 100% (= 1). To distinguish between the impact of BcrC and UppP on this total phosphatase activity, the relative contribution of BcrC to the overall phosphatase activity in the absence of bacitracin is quantified by x^{BcrC} and consequently, the relative contribution of UppP is given by $(1-x^{\text{BcrC}})$. This distinction is necessary to describe the increased phosphatase activity that results from higher BcrC levels in response to bacitracin correctly. Thus, in our model, a contribution of 63% of BcrC to the overall phosphatase activity (without bacitracin) involves a relative contribution of 37% of UppP to the overall phosphatase activity. Consequently, $x^{\text{BcrC}} = 0$ would imply that the contribution of BcrC to the overall phosphatase activity is negligible and that the total phosphatase activity is not affected by bacitracin. In contrast, if $x^{\text{BcrC}} = 1$, the total phosphatase activity would be exclusively determined by BcrC and increases in response to bacitracin as BcrC levels increase. Since both phosphatases are assumed to impact the total phosphatase activity, this is given by the weighted sum of the contributions from BcrC and UppP.

The phosphatase activity/expression of UppP is expected to be independent from bacitracin and also invariant in a $\Delta bcrC$ mutant, as suggested by previous experiments (Cao and Helmann, 2002; Zhao *et al.*, 2016; Radeck *et al.*, 2017b) and our own qRT-PCR data (Figure R4).

To clarify the meaning of x^{BcrC} , we revised the manuscript as mentioned above (lines 274-285).

2- Figure 2B panel (I). This figure is confusing since the red and blue curves (where BcrC is knocked down) show higher LUX expression than the black and green curves. Further, it is unclear why the wild type cells start out with higher levels of BcrC compared to the strain where BceAB is knocked-out.

Although little is known about the precise input stimulus of the P_{bcrC} regulator σ^M , it is conceivable that the overproduction of lipid II cycle intermediates in the *bcrC* mutant reported here leads to a higher demand of UPP dephosphorylation, which in turn may signal an increased demand for *bcrC* expression (this could explain the increased basal LUX expression of the red and blue curves).

However, as discussed in response to reviewer #1's second point, we do not feel that the slightly lower levels of the P_{bcrC} activity in a *bceAB* mutant compared to the wildtype is biologically meaningful, given the large error bars of the wildtype measurement in this case. Accordingly, we here assume the same phosphatase activities for WT and *bceAB* mutant also in the mathematical model (see Supplementary text).

3- Figure 5C. Here, while the MIC values match the experimental values, the curve trends seem suspicious, unless backed up by experimental data. In particular, it is strange that the wild-type cells (black curve) and the *bceAB*-knockdown mutant cells show an increase in PG synthesis rate as the drug dose is increased within some range. Please clarify if this overshoot is observed experimentally, or whether it is just an artifact of the model, in which case, the model's predictive capacity is in question.

The curves shown in Figure 5C are the model predictions of the rate of PG synthesis in response to bacitracin treatment. The increased PG synthesis rate for higher bacitracin concentrations is predicted by the model in both strains in which the BcrC resistance module is intact. The induction of *bcrC* expression in response to increasing bacitracin concentrations (Fig. 3A) results in an increased rate of UPP dephosphorylation and therefore in low concentrations of UPP and higher concentrations of all other lipid II cycle intermediates, also lipid II. In the model, this increased concentration of lipid II is the origin of the observed 'overshoot' of the rate of PG synthesis. For sufficiently high concentrations of bacitracin, a significant portion of UPP is bound in an UPP-BAC complex, such that the overall concentration of lipid II cycle intermediates is also significantly reduced and the concentration of lipid II (and all other lipid II cycle intermediates different from UPP) is no longer increasing despite high BcrC levels.

As it is not trivial to measure the rate of PG synthesis experimentally (see explanation in the response to the first comment of reviewer #1), we were not able to test this model prediction experimentally. However, as higher concentrations of bacitracin significantly interfere with the lipid II cycle and thereby disbalance the distribution of lipid II cycle intermediates (as shown in detail in Piepenbreier et al., 2019), it is conceivable that such an 'overshoot' in the PG synthesis rate could make some biological sense, as the cell can use this to protect itself from the antibiotic. This is indeed predicted by the model, as evident from increased IC50 values in *bcrC*⁺ vs. *bcrC*⁻ strains (Fig. 6B).

4- In the subsection "Calibration of the mathematical model" of the Supplementary text, it is not clear which observations were made experimentally, and which were seen in the model. Please state these clearly.

We thank the reviewer for this important comment and revised the whole paragraph to make the distinction between experiment and model more obvious:

"Calibration of the mathematical model"

In order to calibrate the model, we aimed to identify physiologically relevant values for the parameters in Eqs. (I-XIV). At first, we set all known parameters from the model of the lipid II cycle to its previously defined values (see Supplementary Table 3).

Subsequently, we determined the new parameters arising from the mathematical description of the BcrC resistance module, namely x^{BcrC} , which defined s^{BcrC} , and s^{UPP} , respectively, within the model. For this purpose, we compared the scenarios of a strain lacking both resistance modules ($\Delta bceAB\Delta bcrC$ mutant) or featuring the BcrC resistance module solely ($\Delta bceAB$ mutant) from a theoretical point of view: Each of the two parameters affects both

the progression of the lipid II cycle without bacitracin and the effect of bacitracin treatment on the lipid II cycle. While a higher impact of BcrC on the overall phosphatase activity supports the progression of the cycle more efficiently when BcrC is present, the rate of PG synthesis would be reduced more strongly in this case when BcrC is lacking. This demands a more pronounced upregulation of the production of lipid II cycle intermediates in response to *bcrC* deletion to recover a close-to-optimal PG synthesis rate. However, the model also predicted a PG synthesis rate above the optimal one when assuming an excessive upregulation of lipid carrier production, which is not valid in a physiological sense. Furthermore, variations in the PG synthesis rate without bacitracin treatment clearly imply significant differences in the amount of bacitracin cells can stand. Obviously, in the scenario where BcrC is present as a resistance module ($\Delta bceAB$ mutant), a stronger contribution of BcrC to the overall phosphatase activity confers higher resistance and coincides with a raised IC_{50} . However, when lacking BcrC ($\Delta bceAB\Delta bcrC$ mutant), the PG synthesis rate without bacitracin treatment dictates the susceptibility towards bacitracin. If the PG synthesis rate is still distinctly affected in the untreated scenario, little amounts of bacitracin would be sufficient to reduce the PG synthesis rate to half of its optimum. In contrast, much higher bacitracin concentrations are required to reach 50% of the optimal PG synthesis rate when the rate is nearly unaffected without antibiotic. Thus, as the PG synthesis rate without bacitracin treatment is governed by the contribution of BcrC on the overall phosphatase activity (x^{BcrC}) and lipid carrier upregulation in response to BcrC shortage (s^{UPP}) – as explained above – the IC_{50} prediction of the model for the $\Delta bceAB\Delta bcrC$ mutant strongly depends on these two parameters. We ultimately aimed to find a theoretical model that simultaneously describe the progression of the lipid II cycle with ($\Delta bceAB$) and without BcrC ($\Delta bcrC\Delta bceAB$) precisely and matches physiological conditions as well. Therefore, we simulated the IC_{50} model predictions for 50x50 combinations of the two parameters x^{BcrC} and s^{UPP} and determined the weighted squared 2-norm χ^2 for all possible combinations as follows:

$$\chi^2(x^{BcrC}, s^{UPP}) = \frac{(IC_{50}^{\Delta bceAB}(x^{BcrC}, s^{UPP}) - MIC^{\Delta bceAB})^2}{(\sigma_{MIC^{\Delta bceAB}})^2} + \frac{(IC_{50}^{\Delta bceAB\Delta bcrC}(x^{BcrC}, s^{UPP}) - MIC^{\Delta bceAB\Delta bcrC})^2}{(\sigma_{MIC^{\Delta bceAB\Delta bcrC}})^2}$$

Here, $MIC^{\Delta bceAB}$ and $MIC^{\Delta bceAB\Delta bcrC}$ represent the experimentally determined MICs in the different strains and $\sigma_{MIC^{\Delta bceAB}}$ and $\sigma_{MIC^{\Delta bceAB\Delta bcrC}}$ the respective errors in the experimental MICs, calculated by the error propagation formula (errors are given in the main text). Furthermore, $IC_{50}^{\Delta bceAB}$ and $IC_{50}^{\Delta bceAB\Delta bcrC}$ describe the model-predicted IC_{50} 's, dependent on the parameters x^{BcrC} and s^{UPP} .

To find the optimal parameter combination, we demanded the following two constraints for the two model parameters:

i. $\chi^2(x^{BcrC}, s^{UPP}) \rightarrow \min$

and

ii. $j_{PG}^{\Delta bceAB\Delta bcrC}(x^{BcrC}, s^{UPP}) \leq j_{PG}^{WT}$
 j_{PG}^{WT} and $j_{PG}^{\Delta bceAB\Delta bcrC}$ display the theoretical rates of PG synthesis without bacitracin treatment in the wild-type scenario and a scenario where both resistance modules are lacking, respectively. The second constraint accounts for the physiological plausible limitation of the PG synthesis rate to its wild-type level. In Supplementary Figure S3, the χ^2 values are plotted against the parameter combinations. Standard deviations on the two model parameters were determined from the 68.3 % confidence intervals as described (Press *et al.*, 1992) and also illustrated in Supplementary Figure S3. The final parameter values of x^{BcrC} and s^{UPP} as well as their standard deviation σ are given in Supplementary Table 3).

Finally, we determined the parameters originating from the previous model of the BceAB resistance module. Since the setup of the experimental measurements of the expression levels of the resistance modules, which we now aimed to quantitatively describe by the new model, was quite different from the previous experimental study that was used to calibrate the pre-existing model, we were not able to transfer the existing parameters to our new

model. Rather, significant variations in the growth conditions between both experimental approaches demanded adaptations in the model parameters that describe the dynamics of the Bce system theoretically. Therefore, we fixed the model parameters that are independent from the growth conditions (e.g. mRNA degradation rates, translation rate) to their pre-defined, physiological values and determined the remaining ones by a constrained optimization approach. Here, the experimental data of the P_{bceAB} -*luxABCDE* reporter output provide nine objectives to the seven unknown model parameters. To solve this over-determined non-linear data-fitting problem, we used the solving function *lsqnonlin*, embedded in the MATLAB™ software. This function solves nonlinear least-square curve fitting problems of the form

$$\min \|f(x)\|_2^2 = \min (f_1(x)^2 + f_2(x)^2 + \dots + f_n(x)^2)$$

by using a trust-region reflective Newton algorithm. As outputs, it returns the optimal parameter set \bar{x} of the problem as well as the squared-2 norm χ^2 of the residual at \bar{x} ($\chi^2 = \sum f(\bar{x})^2$). To account for the presence of local optima, 50 independent fits were performed with randomly chosen initial parameter sets and the best-fit result was given at minimal χ^2 . The optimal parameters are shown in Supplementary Table S3. We followed (Wall *et al.*, 2009) to compute the asymmetric errors σ_+ and σ_- with respect to the optimal parameter values \bar{x} , listed in Supplementary Table S3. The squared errors for the parameter x_k were calculated using the following equations:

$$\sigma_{k,+}^2 = \frac{\sum_{i:x_{k,i} > \bar{x}_k} (x_{k,i} - \bar{x}_k)^2 e^{-\chi_i^2/2}}{\sum_{i:x_{k,i} > \bar{x}_k} e^{-\chi_i^2/2}}$$

and

$$\sigma_{k,-}^2 = \frac{\sum_{i:x_{k,i} < \bar{x}_k} (x_{k,i} - \bar{x}_k)^2 e^{-\chi_i^2/2}}{\sum_{i:x_{k,i} < \bar{x}_k} e^{-\chi_i^2/2}}$$

Where $x_{k,i}$ is the value of the parameter x_k in the i^{th} fit, \bar{x}_k is the value of x_k in the fit with the lowest value of χ^2 , and χ_i^2 is the value of χ^2 for the i^{th} fit. In using the likelihood function $e^{-\chi_i^2/2}$, we assumed that the errors in the measurements are independent and normally distributed with widths equal to the standard error of the mean.

”

December 19, 2019

Dr. Georg Fritz
The University of Western Australia
School of Molecular Sciences
Perth 6009
Australia

Re: mSystems00687-19 (From modules to networks: A systems-level analysis of the bacitracin stress response in *Bacillus subtilis*)

Dear Dr. Georg Fritz:

Please, address the comments of Reviewer 1.

Below you will find the comments of the reviewers.

To submit your modified manuscript, log onto the eJP submission site at <https://msystems.msubmit.net/cgi-bin/main.plex>. If you cannot remember your password, click the "Can't remember your password?" link and follow the instructions on the screen. Go to Author Tasks and click the appropriate manuscript title to begin the resubmission process. The information that you entered when you first submitted the paper will be displayed. Please update the information as necessary. Provide (1) point-by-point responses to the issues raised by the reviewers as file type "Response to Reviewers," not in your cover letter, and (2) a PDF file that indicates the changes from the original submission (by highlighting or underlining the changes) as file type "Marked Up Manuscript - For Review Only."

Please return the manuscript within 60 days; if you cannot complete the modification within this time period, please contact me. If you do not wish to modify the manuscript and prefer to submit it to another journal, please notify me of your decision immediately so that the manuscript may be formally withdrawn from consideration by mSystems.

To avoid unnecessary delay in publication should your modified manuscript be accepted, it is important that all elements you upload meet the technical requirements for production. I strongly recommend that you check your digital images using the Rapid Inspector tool at <http://rapidinspector.cadmus.com/RapidInspector/zmw/>.

Sincerely,

Domitilla Del Vecchio

Editor, mSystems

Journals Department
Reviewer comments:

Reviewer #1 (Comments for the Author):

The authors have addressed my comments on this manuscript. However, could authors further clarify that from which snapshot point of the temporal OD data in the supplementary Figure S1, the Lux activities (i.e. RLU/OD) in Fig2B(i) and (iii) were calculated? This could also serve as a double-check on the presented data.

According to the description of luciferase assay in the method, bacitracin was added one hour after inoculation at OD₆₀₀=0.01 and the data were recorded every 5 min. The x-axis of the temporal data in the supplementary Figure S1 starts from t=50 min. By counting the data point of every 5 min and by observing a clear jump in OD value between the 4th and 5th data point which is probably due to the operation time of adding bacitracin, could the authors confirm that the first recorded data after adding bacitracin is the 5th data point from t=50 min on the x-axis? Because the OD value is less than 1 and is in the denominator, the ratio RLU/OD would be significantly affected by a slight change of the OD value. The difference between the green and black curves in Fig2B(i) and a lower RLU/OD at the point (bacitracin=0.03 ug/mL) of the black curve in Fig2B(iii) may result from mistaking the temporal data points before and after a jump of the OD values. The extents of the indicated differences are similar to a jump of OD value between the 4th and 5th points in the temporal data in the supplementary Figure S1. For instance, in the P_{becAB}-WT subplot of the supplementary Figure S1, the OD values for bacitracin=0~10 ug/mL looks similar after the 6th data point counting from t=50 min and assuming luciferase level are similar for low bacitracin=0~0.03 ug/mL (because the data were not shown), it doesn't make sense that the RLU/OD of bacitracin=0.03 ug/mL is lower than the ones of bacitracin=0 and 0.01 ug/mL. Could the authors double check which temporal data you picked?

Reviewer #2 (Comments for the Author):

All previous comments have been addressed.

Reviewer #1 (Comments for the Author):

The authors have addressed my comments on this manuscript. However, could authors further clarify that from which snapshot point of the temporal OD data in the supplementary Figure S1, the Lux activities (i.e. RLU/OD) in Fig2B(i) and (iii) were calculated? This could also serve as a double-check on the presented data.

According to the description of luciferase assay in the method, bacitracin was added one hour after inoculation at OD600=0.01 and the data were recorded every 5 min. The x-axis of the temporal data in the supplementary Figure S1 starts from t=50 min. By counting the data point of every 5 min and by observing a clear jump in OD value between the 4th and 5th data point which is probably due to the operation time of adding bacitracin, could the authors confirm that the first recorded data after adding bacitracin is the 5th data point from t=50 min on the x-axis?

We are very grateful to the reviewer for her/his detailed inspection of our data. We indeed noticed that we mistakenly wrote in the Methods section that measurements were taken every 5 min, although we actually took them every 3 min. This is now corrected in the revised version of the manuscript and should explain the confusion the reviewer had with this data. In Figure R1, which is an excerpt from Supplementary Figure S1, we provide a detailed annotation of the timing of the data points, together with an annotation of the timepoint at which the data was taken for the dose-response curves in Fig. 2B(i) and (iii), namely 60 min after induction, i.e. at time 120 min.

Figure R1. Detailed annotation of timepoints in Supplementary Figure S1. Bacitracin was added right before timepoint $t = 60.3$ min. The analysis of luminescence data was performed at time 120.6 min, i.e. 1 hour after induction with bacitracin. Here it is visible that at the time of evaluation (120.6 min), the OD values are almost identical between wildtype (WT) and $\Delta bceAB$ mutant (OD = 0.11).

Because the OD value is less than 1 and is in the denominator, the ratio RLU/OD would be significantly affected by a slight change of the OD value. The difference between the green and black curves in Fig2B(i) and a lower RLU/OD at the point (bacitracin=0.03 ug/mL) of the black curve in Fig2B(iii) may result from mistaking the temporal data points before and after a jump of the OD values. The extents of the indicated differences are similar to a jump of OD value between the 4th and 5th points in the temporal data in the supplementary Figure S1. For instance, in the P_{becAB}-WT subplot of the supplementary Figure S1, the OD values for bacitracin=0~10 ug/mL looks similar after the 6th data point counting from t=50 min and assuming luciferase level are similar for low bacitracin=0~0.03 ug/mL (because the data were not shown), it doesn't make sense that the RLU/OD of bacitracin=0.03 ug/mL is lower than the ones of bacitracin=0 and 0.01 ug/mL. Could the authors double check which temporal data you picked?

As the reviewer correctly notes there is the jump in OD data at t = 60.3 min after bacitracin addition, which arises from the fact that cells keep growing while they are out of the reader for bacitracin addition. However, as stated above and as described in the Methods section, the analysis of the bacitracin response of P_{bcrC} and P_{bceAB} in Fig. 2B(i) and (iii) was performed 1 hour *after* bacitracin addition, i.e. at t = 120 min in **Fig. R1**. At this time the OD values of wildtype (WT) and $\Delta bceAB$ strains are identical (OD ~ 0.11), indicating that the hypothesis of the reviewer is not the explanation for the difference between the lower P_{bcrC} activity in the $\Delta bceAB$ mutant compared to the WT. This is further substantiated by **Fig. R2**, clearly showing that at t = 120 min OD is identical between WT and $\Delta bceAB$ mutant, while normalized luminescence (RLU/OD) of P_{bcrC}-lux is higher for the wildtype. Thus, the data shown in Figure 2B(i) is not flawed by fluctuations in the OD values.

Figure R2. Growth (top) and normalized luminescence of the P_{bcrC}-lux reporter (bottom) as a function of time in minutes (x-axis). Comparison shows data of wildtype and $\Delta bceAB$ mutant in the absence of bacitracin. Data represents the mean of 3 independent biological replicates.

Similarly, the slightly lower activity of the P_{bceAB}-WT reporter at 0.03 ug/mL is within the error bar to the 0 ug/mL and 0.01 ug/ml data points (Fig. 2B(iii)), and thus, not significantly different. This is also evident when studying the time course of the luciferase reporter, as requested by the reviewer (**Fig. R3**; bottom). Here you see that at 0.03 ug/mL the data point used for the analysis in Fig. 2B(iii) (t=120min; highlighted by the red arrow) is a mere fluctuation and does not reflect a systematically lower promoter activity at this bacitracin concentration.

We thank the reviewer again for her/his critical inspection of our data, which has helped to eliminate errors and has significantly improved the manuscript!

Figure R3. Growth (top) and normalized luminescence of the P_{bceA}-lux (WT) reporter (bottom) as a function of time in minutes (x-axis). The three curves show the behavior when cells were treated with bacitracin at t = 60 min. Here it is clearly visible that the data point under 0.03 ug/mL bac at 1 hour post-induction (t = 120 min) is a mere fluctuation (red arrow) and not a systematic trend, as suspected by the reviewer. Data represents the mean of 3 independent biological replicates. Note that the high fluctuations of the data between t = 50 min and 100 min arise from the fact that the luminescence signal is close to the detection limit of the plate reader, and division by low OD values amplifies the measurement noise within this regime.

December 28, 2019

Dr. Georg Fritz
The University of Western Australia
School of Molecular Sciences
Perth 6009
Australia

Re: mSystems00687-19R1 (From modules to networks: A systems-level analysis of the bacitracin stress response in *Bacillus subtilis*)

Dear Dr. Georg Fritz:

Your manuscript has been accepted, and I am forwarding it to the ASM Journals Department for publication. For your reference, ASM Journals' address is given below. Before it can be scheduled for publication, your manuscript will be checked by the mSystems production editor, Ellie Ghatineh, to make sure that all elements meet the technical requirements for publication. She will contact you if anything needs to be revised before copyediting and production can begin. Otherwise, you will be notified when your proofs are ready to be viewed.

Sincerely,

Domitilla Del Vecchio
Editor, mSystems

Journals Department
Supplementary Table S3: Accept
Supplementary Figure S1: Accept
Supplementary Table S1: Accept
Supplementary Text 1: Accept
Supplementary Figure S2: Accept
Supplementary Table S2: Accept